# Homeostatic regulation of rapid eye movement sleep by the preoptic area of the hypothalamus

John J Maurer[1], Alexandra Lin[1], Xi Jin[1], Jiso Hong[1], Nicholas Sathi[1], Romain Cardis[2], Alejandro Osorio-Forero[2], Anita Lüthi[2], Franz Weber[1], Shinjae Chung[1]*

[1]Department of Neuroscience, Chronobiology and Sleep Institute, Perelman School of Medicine, University of Pennsylvania, Philadelphia, United States; [2]Department of Fundamental Neurosciences, University of Lausanne, Lausanne, Switzerland

*For correspondence:
shinjaec@pennmedicine.upenn.edu

Competing interest: The authors declare that no competing interests exist.

**Abstract** Rapid eye movement sleep (REMs) is characterized by activated electroencephalogram (EEG) and muscle atonia, accompanied by vivid dreams. REMs is homeostatically regulated, ensuring that any loss of REMs is compensated by a subsequent increase in its amount. However, the neural mechanisms underlying the homeostatic control of REMs are largely unknown. Here, we show that GABAergic neurons in the preoptic area of the hypothalamus projecting to the tuberomammillary nucleus (POA$^{GAD2}$→TMN neurons) are crucial for the homeostatic regulation of REMs in mice. POA$^{GAD2}$→TMN neurons are most active during REMs, and inhibiting them specifically decreases REMs. REMs restriction leads to an increased number and amplitude of calcium transients in POA$^{GAD2}$→TMN neurons, reflecting the accumulation of REMs pressure. Inhibiting POA$^{GAD2}$→TMN neurons during REMs restriction blocked the subsequent rebound of REMs. Our findings reveal a hypothalamic circuit whose activity mirrors the buildup of homeostatic REMs pressure during restriction and that is required for the ensuing rebound in REMs.

## eLife assessment

This **valuable** study advances our understanding of the brain nuclei involved in rapid-eye movement (REM) sleep regulation. Using a combination of imaging, electrophysiology, and optogenetic tools, the study provides **convincing** evidence that inhibitory neurons in the preoptic area of the hypothalamus influence REM sleep. This work will be of interest to neurobiologists working on the brain circuits of sleep.

## Introduction

Rapid eye movement sleep (REMs) is homeostatically regulated as demonstrated in various species including mice, rats, cats, and humans (*Siegel and Gordon, 1965*; *Beersma et al., 1990*; *Benington et al., 1994*; *Endo et al., 1997*; *Endo et al., 1998*; *Rechtschaffen et al., 1999*; *Franken, 2002*; *Shea et al., 2008*). REMs restriction leads to an increased homeostatic need for REMs. During the subsequent recovery sleep, the amount of REMs is increased to compensate for the amount lost during restriction. While seminal dissection studies indicate a pontine origin of REMs, recent studies have revealed that neural populations in the hypothalamus, midbrain, amygdala and medulla regulate REMs by promoting or inhibiting REMs (*Clément et al., 2011*; *Jego et al., 2013*; *Hayashi et al., 2015*; *Van Dort et al., 2015*; *Weber et al., 2015*; *Weber et al., 2018*; *Chung et al., 2017*; *Torontali*

*et al., 2019*). However, we do not have a clear understanding about the homeostatic mechanisms regulating REMs and which brain regions integrate homeostatic REMs pressure.

The preoptic area of the hypothalamus (POA) is crucial for sleep regulation. The POA contains neurons that become activated during sleep and that are sufficient and necessary for sleep (*Economo, 1930*; *Nauta, 1946*; *McGinty and Sterman, 1968*; *Sallanon et al., 1989*; *Sherin et al., 1996*; *Szymusiak et al., 1998*; *Lu et al., 2000*; *Gong et al., 2004*; *Takahashi et al., 2009*; *Zhang et al., 2015*; *Kroeger et al., 2018*). Specifically, POA GABAergic neurons projecting to the tuberomammillary nucleus (POA$^{GAD2}$→TMN neurons) form a subpopulation of sleep-active POA neurons, and promote sleep when optogenetically activated (*Chung et al., 2017*). A previous study demonstrated that c-Fos expression in the POA is increased in REMs restricted rats (*Gvilia et al., 2006*). However, the molecular identity of POA neurons that become activated during heightened REMs pressure and whether their activity is necessary for homeostatic REMs regulation remains largely unknown.

In this study, using fiber photometry, we found that POA$^{GAD2}$→TMN neurons become gradually activated during non-rapid eye movement sleep (NREMs) before the onset of REMs and are most active during REMs. Optogenetic inhibition of POA$^{GAD2}$→TMN neurons significantly decreased REMs. We therefore hypothesized that the POA$^{GAD2}$→TMN neurons are well suited to encode homeostatic pressure for REMs. Using fiber photometry recordings combined with REMs restriction, we show that the activity of POA$^{GAD2}$→TMN neurons significantly increased during periods of heightened REMs pressure. Optogenetic inhibition of POA$^{GAD2}$→TMN neurons during REMs restriction prevented the subsequent increase of REMs during recovery sleep. Our findings identify a hypothalamic circuit regulating the homeostatic need for REMs.

## Results

### POA$^{GAD2}$→TMN neurons are most active during REMs

To monitor the population activity of POA$^{GAD2}$→TMN neurons in vivo during spontaneous sleep, we performed fiber photometry recordings. GAD2-Cre mice were injected with retrograde adeno-associated viruses (AAVs) encoding Cre-inducible GCaMP8s (AAVretro-FLEX-jGCaMP8s) into the TMN (*Tervo et al., 2016*; *Zhang et al., 2023*), and an optic fiber was implanted into the POA (*Figure 1A*; virus expression and optic fiber tracts were located in the ventrolateral POA, lateral POA, and the lateral part of medial POA). The calcium activity of POA$^{GAD2}$→TMN neurons was significantly higher during REMs compared with that during wake and NREMs (*Figure 1B and C*; detailed statistical results are shown in Figure legends and *Supplementary file 1*).

We further analyzed the activity changes of the POA$^{GAD2}$→TMN neurons during NREMs→REMs or NREMs→wake transitions. We found that the calcium activity of POA$^{GAD2}$→TMN neurons during NREMs→ REMs transitions becomes significantly increased 40 s before the REMs onset and remains elevated throughout REMs (*Figure 1D*). During NREMs→wake transitions, the activity of POA$^{GAD2}$→TMN neurons started rising 10 s before the wake onset and remained elevated for 10 s after the onset (*Figure 1D*). The ΔF/F activity gradually increased throughout NREMs episodes (*Figure 1E*).

Given that both the θ and σ (6–9 and 10.5–16 Hz) power increased preceding the REMs onset (*Figure 1D*), we investigated the relationship between the POA$^{GAD2}$→TMN neuron activity and the spectral composition of the EEG in more detail (*Figure 1F*). We used a linear regression model to predict the current POA$^{GAD2}$→TMN neural activity based on the preceding and following spectral EEG features (*Weber et al., 2010*; *Schott et al., 2023*). We found that the activity of POA$^{GAD2}$→TMN neurons is preceded by an increase in the θ and σ power and reduction in the δ (0.5–4.5 Hz) power (*Figure 1F*). These changes in the EEG are characteristic for the stage of NREMs preceding REMs (*Gottesmann, 1996*) and their correlation with the activity of POA$^{GAD2}$→TMN neurons is consistent with a role of these neurons in promoting NREMs to REMs transitions.

In a complementary experiment, we monitored the calcium activity of POA$^{GAD2}$→TMN axonal fibers. GAD2-Cre mice were injected with AAV-FLEX-GCaMP6s into the POA and an optic fiber was implanted into the TMN (*Figure 1—figure supplement 1A*). Similar to the activity found for POA$^{GAD2}$→TMN neurons using retrograde AAVs (*Figure 1B and C*), POA$^{GAD2}$→TMN fibers were most active during REMs (*Figure 1—figure supplement 1B and C*). During NREMs, the activity of POA$^{GAD2}$→TMN axonal fibers gradually increased before transitioning to REMs (*Figure 1—figure supplement 1D*), demonstrating that the activity of POA$^{GAD2}$→TMN axonal fibers closely resembles that of cell bodies.

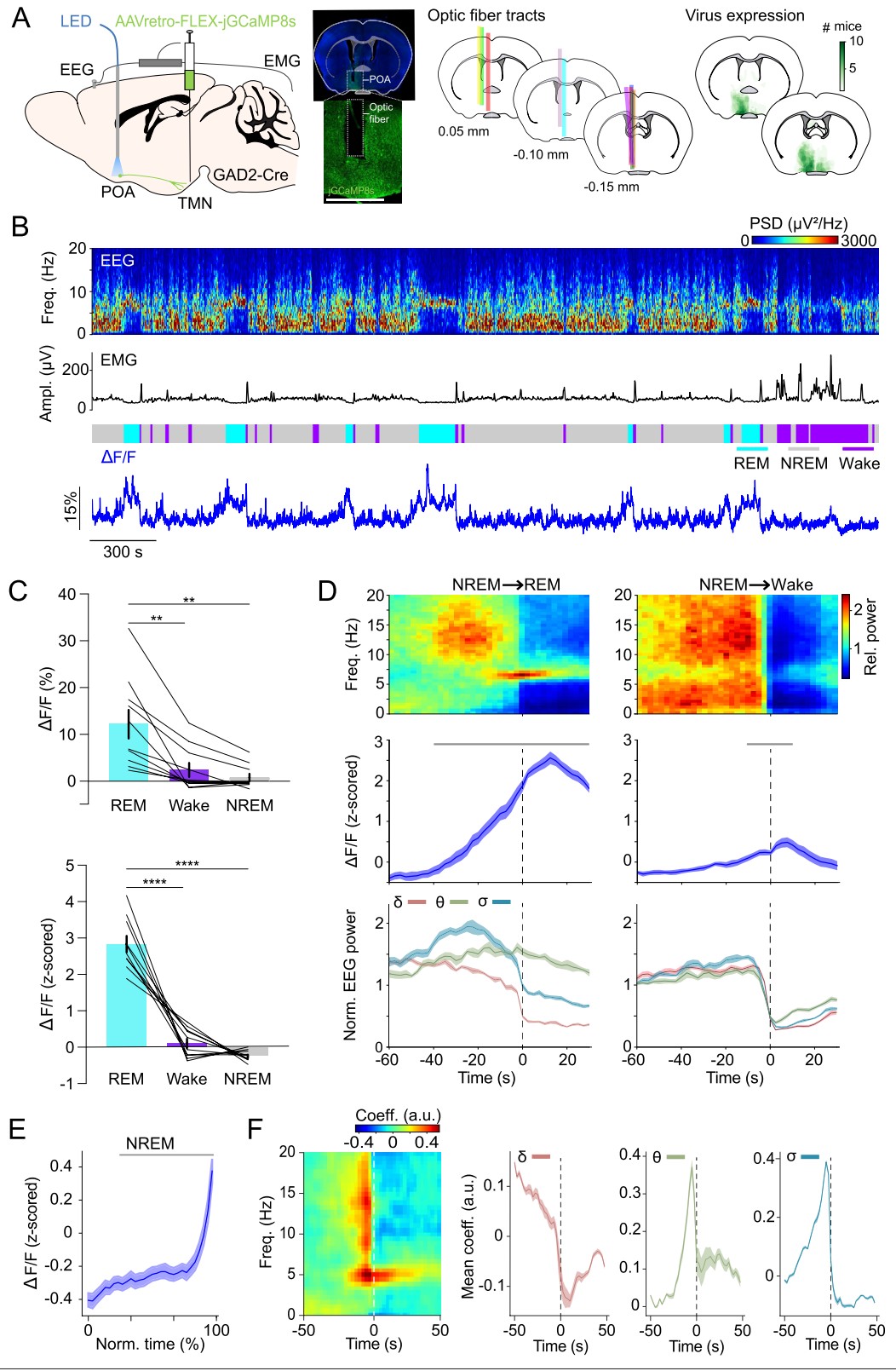

**Figure 1.** POA^GAD2→TMN neurons are most active during rapid eye movement sleep (REMs). (**A**) Left, schematic of fiber photometry with simultaneous electroencephalogram (EEG) and electromyogram (EMG) recordings. Mouse brain figure adapted from the Allen Reference Atlas - Mouse Brain. Center left, fluorescence image of POA in a GAD2-Cre mouse injected with AAVretro-FLEX-jGCaMP8s into the TMN. Scale bar, 1 mm. Center right, location

*Figure 1 continued on next page*

*Figure 1 continued*

of fiber tracts. Each colored bar represents the location of optic fibers for photometry recordings. Right, heatmaps outlining areas with cell bodies expressing GCaMP8. The green color code depicts how many mice the virus expression overlapped at the corresponding location (n=10 mice). (**B**) Example fiber photometry recording. Shown are EEG spectrogram, EMG amplitude, color-coded brain states, and ΔF/F signal. (**C**) Non-normalized and z-scored ΔF/F activity during REMs, wake, and non-rapid eye movement sleep (NREMs). Bars, averages across mice; lines, individual mice; error bars, ± s.e.m. One-way repeated measures (rm) ANOVA, p=9e-4, 2e-6 for non-normalized ΔF/F and z-scored ΔF/F signals; pairwise t-tests with Bonferroni correction, non-normalized ΔF/F, p=0.0056, 0.0039 for REMs vs. wake and REMs vs. NREMs; z-scored ΔF/F, p=6e-5, 1e-7. n=10 mice. (**D**) Average EEG spectrogram (top), z-scored ΔF/F activity (middle) and normalized EEG δ, θ, and σ power (bottom) during NREMs→REMs transitions (left) and NREMs→wake transitions (right). Shading, ± s.e.m. One-way rm ANOVA, p=3.18e-49, 9.50e-8 for NREMs→REMs and NREMs→wake; pairwise t-tests with Holm-Bonferroni correction, NREMs→REMs p<0.0419 between –40 and 30 s, NREMs→wake p<0.0106 between –10 and 10 s. Gray bar, period when ΔF/F activity was significantly different from baseline (–60 to –50 s). n=10 mice. (**E**) ΔF/F activity during NREMs. The duration of NREMs episodes was normalized in time, ranging from 0% to 100%. Shading, ± s.e.m. Pairwise t-tests with Holm-Bonferroni correction p<8.14e-9 between 20 and 100. Gray bar, intervals where ΔF/F activity was significantly different from baseline (0% to 20%, the first time bin). n=566 events. (**F**) Left, linear filter mapping the normalized EEG spectrogram onto the POA$^{GAD2}$→TMN neural activity. Time point 0 s corresponds to the predicted neural activity. Right, coefficients of the linear filter for δ, θ, and σ power band. Shading, ± s.e.m. See ***Supplementary file 1*** for the actual p-values.

The online version of this article includes the following figure supplement(s) for figure 1:

**Figure supplement 1.** POA$^{GAD2}$→TMN axonal fibers are most active during rapid eye movement sleep (REMs).

**Figure supplement 2.** TMN$^{HIS}$ neurons are least active during sleep.

In summary, these results demonstrate that the activity in POA$^{GAD2}$→TMN neurons increases prior to the onset of REMs episodes, following an increase in the θ and σ power in the EEG.

The TMN contains histamine neurons (TMN$^{HIS}$), and previous studies showed that POA$^{GAD2}$ neurons innervate TMN$^{HIS}$ neurons (*Chung et al., 2017*; *Saito et al., 2018*). Consistent with previous electrophysiological studies (*Steininger et al., 1999*; *Vanni-Mercier et al., 2003*; *John et al., 2004*; *Takahashi et al., 2006*), we found in photometry recordings that TMN$^{HIS}$ neurons are highly active during wake and less active during NREMs and REMs (*Figure 1—figure supplement 2A–C*). Consistent with this, the activity of TMN$^{HIS}$ neurons became significantly activated after the transition from NREMs or REMs to wakefulness (*Figure 1—figure supplement 2D*). The TMN$^{HIS}$ neuron activity gradually decreased during NREMs episodes (*Figure 1—figure supplement 2E*), possibly as a result of inhibitory inputs from POA$^{GAD2}$→TMN neurons (*Chung et al., 2017*). Moreover, examining the time course of TMN$^{HIS}$ neurons between two successive REMs episodes (inter-REM interval), we found that their activity gradually decreases throughout the inter-REM interval, reaching its lowest level at the onset of REMs (*Figure 1—figure supplement 2F*). This finding suggests that a minimal activity of TMN$^{HIS}$ neurons is required for entering REMs, and suppression of TMN$^{HIS}$ activity therefore likely facilitates transitions to REMs.

## Inhibiting POA$^{GAD2}$→TMN neurons reduces REMs

To examine whether POA$^{GAD2}$→TMN neurons regulate REMs, we optogenetically inhibited these neurons using the bistable chloride channel SwiChR++ (*Berndt et al., 2016*; *Smith et al., 2024*; *Stucynski et al., 2022*). GAD2-Cre mice were bilaterally injected with retrograde AAVs encoding Cre-inducible SwiChR++ (AAVretro-DIO-SwiChR++-eYFP) or eYFP (AAVretro-DIO-eYFP) into the TMN followed by bilateral optic fiber implantation into the POA (*Figure 2A*). We compared SwiChR++ and eYFP recordings with and without laser stimulation (473 nm, 2 s step pulses at 60 s intervals for 3 hr, zeitgeber time [ZT] 2–5). SwiChR++-mediated inhibition of POA$^{GAD2}$→TMN neurons reduced the amount of REMs compared with recordings without laser stimulation in the same mice and eYFP mice with laser stimulation (*Figure 2B and C*). The overall time spent in wake and NREMs was not altered by SwiChR++-mediated inhibition (*Figure 2C*, *Figure 2—figure supplement 1A and B*), suggesting that POA$^{GAD2}$→TMN neuron activity specifically regulates the amount of REMs.

Next, we compared the spectral composition of the EEG throughout recordings with or without laser stimulation in SwiChR++ and eYFP mice. During REMs, SwiChR++-mediated inhibition of POA$^{GAD2}$→TMN neurons increased the δ and σ power in the EEG compared with SwiChR++-without laser

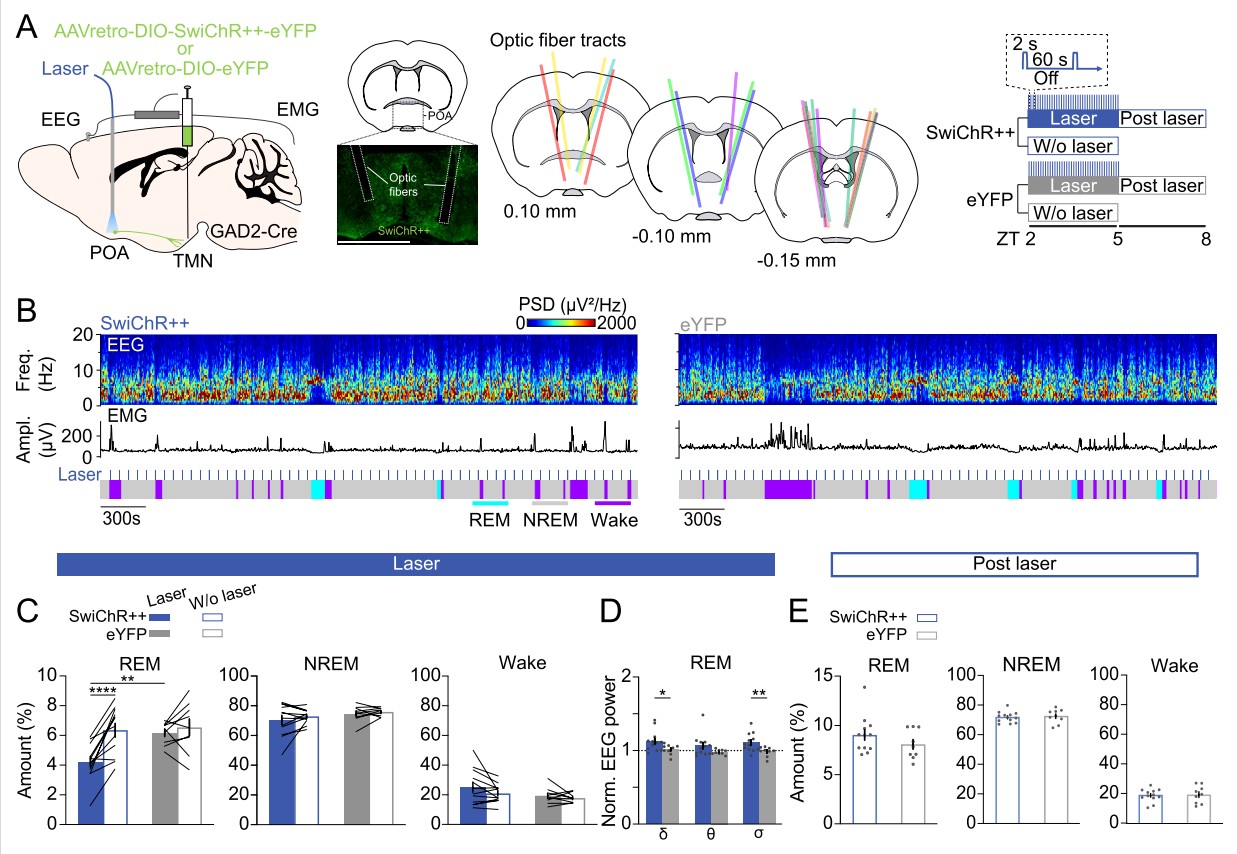

**Figure 2.** Inhibiting POA$^{GAD2}$→TMN neurons reduces rapid eye movement sleep (REMs). (**A**) Left, schematic of optogenetic inhibition experiments. Center left, fluorescence image of POA in a GAD2-CRE mouse injected with AAVretro-DIO-SwiChR++-eYFP into the TMN. Scale bar, 1 mm. Center right, location of optic fiber tracts. Each colored bar represents the location of an optic fiber. Right, experimental paradigm for laser stimulation (2 s step pulses at 60 s intervals) in SwiChR++ and eYFP-expressing mice. Mouse brain figure adapted from the Allen Reference Atlas - Mouse Brain. (**B**) Example recording of a SwiChR++ (left) and eYFP mouse (right) with laser stimulation. Shown are electroencephalogram (EEG) power spectra, EMG amplitude, and color-coded brain states. (**C**) Percentage of time spent in REMs, non-rapid eye movement sleep (NREMs), and wakefulness with and without laser in SwiChR++ and eYFP mice. Mixed ANOVA, virus p=0.0629, laser p=0.0003, interaction p=0.0064; t-tests with Bonferroni correction, SwiChR-laser vs. SwiChR-w/o laser p=3.00e-5, SwiChR-laser vs. eYFP-laser p=0.0058. (**D**) Normalized EEG δ, θ, and σ power during REMs in SwiChR++ and eYFP mice with laser. Unpaired t-tests, SwiChR vs. eYFP p=0.0432, 0.0099 for δ and σ. (**E**) Percentage of time spent in REMs, NREMs, and wakefulness during post-laser sessions. (**C**) Bars, averages across mice; lines, individual mice; error bars, ± s.e.m. (**D, E**) Bars, averages across mice; dots, individual mice; error bars, ± s.e.m. SwiChR++: n=12 mice; eYFP: n=9 mice.

The online version of this article includes the following figure supplement(s) for figure 2:

**Figure supplement 1.** Effects of inhibiting POA$^{GAD2}$→TMN neurons on brain states and electroencephalogram (EEG).

**Figure supplement 2.** Effects of inhibiting TMN$^{HIS}$ neurons on brain states and electroencephalogram (EEG).

and eYFP-laser groups (*Figure 2D*, *Figure 2—figure supplement 1C*). The δ, θ, and σ power in the EEG during NREMs and wake was indistinguishable between SwiChR++ and eYFP groups (*Figure 2— figure supplement 1D*).

Next, we investigated the effect of sustained inhibition of POA$^{GAD2}$→TMN neurons on the following 3 hr recording without laser stimulation (*Figure 2E*). Despite the reduction of REMs during the laser stimulation in SwiChR++ mice (*Figure 2C*), there were no differences in the amount of REMs, NREMs, and wake between SwiChR++ and eYFP groups during the post-laser recordings (*Figure 2E*, *Figure 2—figure supplement 1E and F*), suggesting that the loss of REMs during SwiChR++-mediated inhibition was not followed by a homeostatic increase in REMs.

In a complementary experiment, we also investigated how inhibition of TMN$^{HIS}$ neurons regulates REMs using the same inhibitory optogenetics protocol. HDC-Cre mice were bilaterally injected with AAVs encoding Cre-inducible SwiChR++ (AAV$_2$-EF1a-DIO-SwiChR++-eYFP) or eYFP (AAV$_2$-Ef1α-DIO-eYFP) into the TMN followed by bilateral optic fibers implantation into the TMN (*Figure 2—figure*

*supplement 2A*). Consistent with the previous experiment, we compared SwiChR++ and eYFP recording with and without laser stimulation (*Figure 2—figure supplement 2A and B*). SwiChR++-mediated inhibition of TMN^HIS neurons increased the amount of REMs compared with recordings without laser stimulation in the same mice and eYFP mice with laser stimulation (*Figure 2—figure supplement 2C*). Given that TMN also contains other types of neurons, inhibition of histamine neurons by POA^GAD2→TMN neurons may not be the sole source of the observed effect on REMs upon inhibition of POA^GAD2→TMN neurons.

Taken together, we first demonstrated that SwiChR++-mediated inhibition of POA^GAD2→TMN neurons reduced the amount of REMs, supporting a necessary role of these neurons in REMs regulation. Second, the lost amount of REMs was not compensated for in the subsequent sleep suggesting that their activity is involved in the homeostatic regulation of REMs.

## POA^GAD2→TMN neurons exhibit an increased number of calcium transients during REMs restriction

To probe whether the activity of POA^GAD2→TMN neurons changes during periods of high REMs pressure resulting from REMs restriction, we performed fiber photometry recordings combined with a closed-loop REMs restriction protocol (*Figure 3A*). To detect the onset of REMs, we adapted a previously applied automatic REMs detection algorithm (*Weber et al., 2015*; *Weber et al., 2018*; *Stucynski et al., 2022*; *Schott et al., 2023*). Briefly, the animal's brain state was classified based on real-time analysis of the EEG/EMG signals. As soon as a REMs episode was detected, a small vibrating motor attached to the animal's head was turned on to terminate REMs by briefly awakening the animal (*Figure 3A*, *Figure 3—figure supplement 1A*; *Cardis et al., 2021*; *Osorio-Forero et al., 2023*). Compared with the baseline recordings from the same mice during the same circadian time, we found that 6 hr of REMs restriction (ZT 1.5–7.5) significantly reduced the amount of REMs and duration of REMs episodes, while increasing their frequency (*Figure 3—figure supplement 1A, D-F*; amount: Cohen's d [d]=–1.621; duration: d=–2.669; frequency: d=1.294). The number of motor activations gradually increased as mice tried to enter REMs more frequently, indicating the accumulation of REMs pressure (*Figure 3—figure supplement 1B*). Before the onset of the motor vibration, we found a clear increase in the EEG θ band reflecting NREMs to REMs transitions (*Figure 3—figure supplement 1C*). We also investigated the effects of REMs restriction on the spectral composition of the EEG. We found an increased δ power for REMs during restriction (*Figure 3—figure supplement 1G*). During rebound (ZT 7.5–8.5), the amount of REMs was significantly increased compared with that during the same circadian time due to an increased frequency of REMs episodes (*Figure 3—figure supplement 1H-K*; amount: d=0.868; frequency: d=1.147), compensating for the amount of REMs lost during restriction.

During REMs restriction, we observed an increased number of calcium transients in the activity of POA^GAD2→TMN neurons as the REMs pressure increased (*Figure 3A and B*). To detect and quantify these transients, we applied an algorithm previously applied to detect calcium events in fiber photometry recordings (*Antila et al., 2022*; *Smith et al., 2024*). Using fiber photometry, we monitored the population activity of POA^GAD2→TMN neurons during the last hour of REMs restriction (ZT 6.5–7.5, referred as restr.) and the first hour of REMs rebound (ZT 7.5–8.5, referred as reb.) and compared it with the activity during baseline recordings of the same mice on separate days at the same circadian time (*Figure 3A and B*). To restrict REMs, we used a motor (ZT 1.5–5.5) as previously described or gently pulled a string attached to the animal's head (ZT 5.5–7.5) during the photometry recordings (*Figure 3A*). The manual REMs deprivation was utilized during the last 2 hr of REMs restriction to avoid potential motion artifacts from the vibrating motor that could contaminate calcium signals. We verified that REMs was adequately restricted (*Figure 3—figure supplement 2*). During restriction, the number of calcium transients of the POA^GAD2→TMN neurons was significantly increased compared with that during baseline recordings (*Figure 3C*). In particular, the number of calcium peaks was significantly elevated during NREMs, likely reflecting an increased pressure to transition to REMs (*Figure 3C*). As the duration of REMs episodes was largely reduced as a result of the restriction (*Figure 3—figure supplement 2A*), the number of peaks during REMs was not changed.

We further examined the amplitude of calcium transients during both NREMs and REMs. During REMs restriction, the amplitude of NREMs and REMs calcium transients was significantly higher compared with that during the circadian baseline (*Figure 3D and E*) and remained elevated during NREMs in the rebound phase (*Figure 3F and G*). Overall, these findings show that during periods of

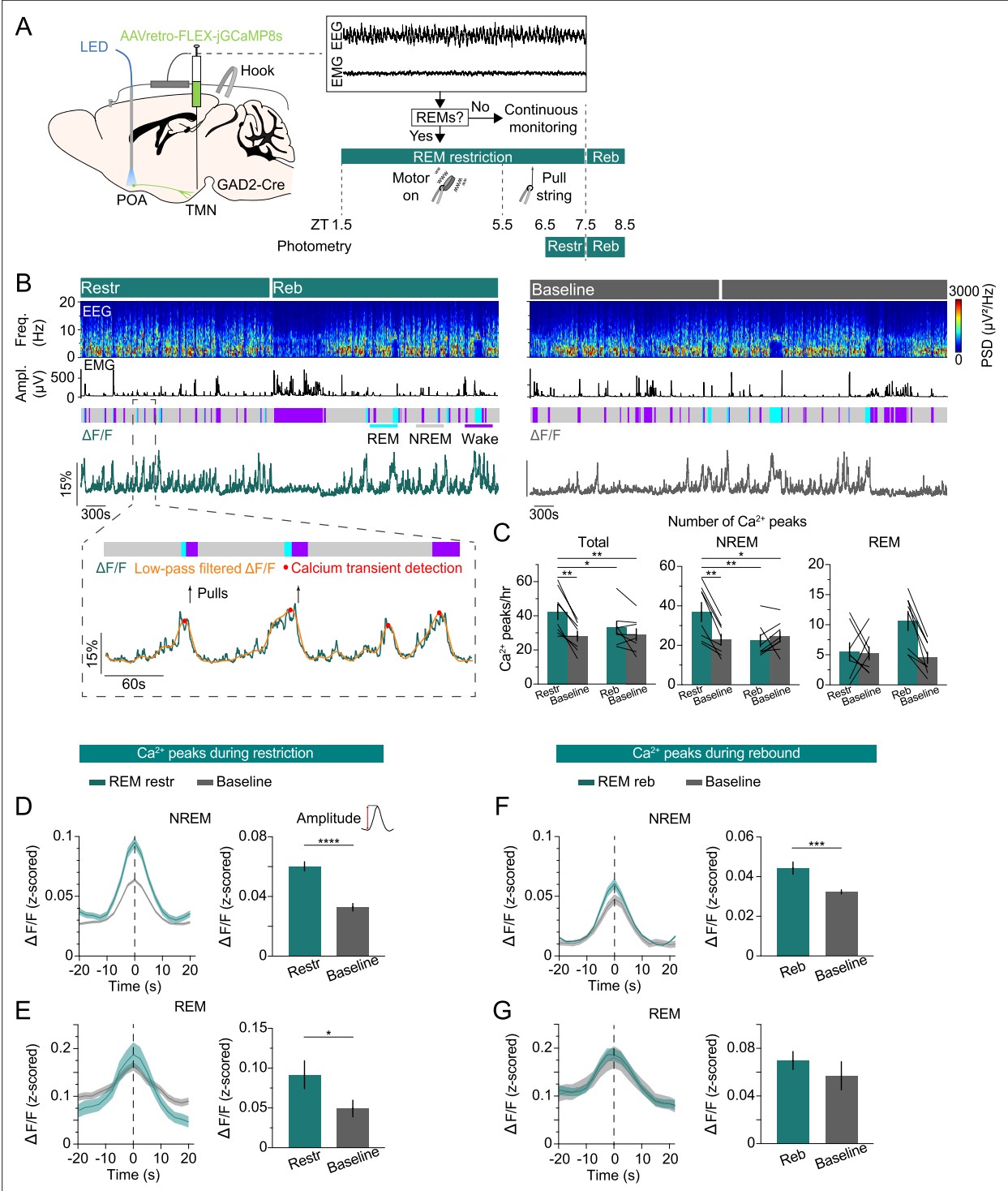

**Figure 3.** POA$^{GAD2}$→TMN neurons exhibit an increased number of calcium transients during rapid eye movement sleep (REMs) restriction. (**A**) Schematic of REMs restriction/rebound and photometry recording experiments. The brain state was continuously monitored; once a REMs episode was detected, we used a vibrating motor (zeitgeber time [ZT] 1.5–5.5) or pulled a string (ZT 5.5–7.5) attached to the mouse head to terminate REMs. Fiber photometry recordings were performed during REMs restriction (ZT 6.5–7.5) and rebound (ZT 7.5–8.5). Mouse brain figure adapted from the Allen Reference Atlas - Mouse Brain. (**B**) Top, example fiber photometry recording. Shown are electroencephalogram (EEG) spectrogram, electromyogram (EMG) amplitude, color-coded brain states, and ΔF/F signal. Bottom, brain states, ΔF/F signal (green), low-pass filtered ΔF/F signal (orange), detected peaks (red), and pulls (arrows) during a selected interval (dashed box) at an expanded timescale. (**C**) Number of calcium peaks during all states (left), non-rapid eye movement sleep (NREMs) (middle), and REMs (right). Bars, averages across mice; lines, individual mice; error bars, ± s.e.m. n=8 mice. Total: two-way

*Figure 3 continued on next page*

*Figure 3 continued*

repeated measures (rm) ANOVA, treatment (baseline vs. manipulation) p=0.0031, time p=0.0613, interaction p=0.0189; t-tests with Bonferroni correction, REM restriction (restr.) vs. baseline (ZT 6.5–7.5) p=0.0033, restr. vs. REM rebound (reb.) p=0.0341, restr. vs. baseline (ZT 7.5–8.5) p=0.0049. NREMs: two-way rm ANOVA, treatment p=0.0233, time p=0.0363, interaction p=0.003; t-tests with Bonferroni correction, restr. vs. baseline (ZT 6.5–7.5) p=0.0057, restr. vs. reb. p=0.0046, restr. vs. baseline (ZT 7.5–8.5) p=0.0115. (D) Left, average NREMs calcium peaks during REMs restriction and baseline recordings. Right, average amplitude of the NREMs calcium peaks. The amplitude was calculated by subtracting the ΔF/F value 10 s before the peak from its value at the peak. Unpaired t-tests, p=6.30e-10. n=295 and 196 peaks during restriction and baseline recordings. (E) Left, average REMs calcium peaks during REMs restriction and baseline recordings. Right, average amplitude of the REMs calcium peaks. Unpaired t-tests, p=0.0437. n=44 and 42 peaks during restriction and baseline recordings. (F) Left, average NREMs calcium peaks during REMs rebound and baseline recordings. Right, average amplitude of the NREMs calcium peaks. Unpaired t-tests, p=0.0002. n=180 and 196 peaks during rebound and baseline recordings. (G) Left, average REMs calcium peaks during REMs rebound and baseline recordings. Right, average amplitude of the REMs calcium peaks. n=84 and 36 peaks during rebound and baseline recordings. (D–G) Bars, averages across trials; error bars, ± s.e.m; shadings, ± s.e.m.

The online version of this article includes the following figure supplement(s) for figure 3:

**Figure supplement 1.** Effects of rapid eye movement sleep (REMs) restriction on brain states and electroencephalogram (EEG).

Related to *Figure 3*. (A) Example session during REMs restriction (top) and REMs rebound (bottom) from the same mouse. Shown are EEG spectrogram, electromyogram (EMG) amplitude, motor vibration events, and color-coded brain states. (B) Frequency of motor vibration events throughout REMs restriction (zeitgeber time [ZT] 1.5–7.5). Error bars, ± s.e.m. One-way ANOVA p=0.0061. (C) Average normalized EEG spectrogram preceding motor vibration onset. (D) Percentage of time spent in REMs, non-rapid eye movement sleep (NREMs), and wakefulness during 6 hr of REMs restriction (green, ZT 1.5–7.5) and baseline recordings (gray, ZT 1.5–7.5). Paired t-test, p=0.0006 for REMs amount. (E) Duration of REMs, NREMs, and wake episodes during 6 hr of REMs restriction (green) and baseline recordings (gray) (ZT 1.5–7.5). Paired t-tests, p=1.44e-5, 0.0276 for the duration of REMs and NREMs episodes. (F) Frequency of REMs, NREMs, and wake episodes during 6 hr of REMs restriction (green) and baseline recordings (gray) (ZT 1.5–7.5). Paired t-tests, p=0.0027 and 0.0043 for the frequency of REMs and NREMs episodes. (G) Power spectral density (PSD) of EEG during REMs, NREMs, and wakefulness during 6 hr of REMs restriction (green) and baseline recordings (gray) (ZT 1.5–7.5). Paired t-tests, p=0.0293 for REMs δ power. (H) Percentage of time spent in REMs, NREMs, and wakefulness during 1 hr of REMs rebound (green, ZT 7.5–8.5) and baseline sleep (gray, ZT 7.5–8.5). Paired t-tests, p=0.0226 for REMs amount. (I) Duration of REMs, NREMs, and wake episodes during 1 hr of REMs rebound (green) and baseline recordings (gray) (ZT 7.5–8.5). (J) Frequency of REMs, NREMs, and wake episodes during 1 hr of REMs rebound (green) and baseline recordings (gray). Paired t-tests, p=0.0055 for the frequency of REMs episodes (ZT 7.5–8.5). (K) PSD of EEG during REMs, NREMs, and wakefulness during 1 hr of REMs rebound (green) and baseline recordings (gray). Paired t-tests, p=0.0164 for wake δ power (ZT 7.5–8.5). (D–F, H–J) Bars, averages across mice; lines, individual mice; error bars, ± s.e.m. n=10 mice. (G, K) Shadings, ± s.e.m. n=10 mice in G; n=9 mice in K, 1 mouse was excluded for REMs because there was no REMs during baseline recordings.

**Figure supplement 2.** Rapid eye movement sleep (REMs) amount, duration, and frequency of episodes during photometry recordings combined with REMs restriction and rebound.

high REMs pressure, the POA$^{GAD2}$→TMN neurons exhibit an increased number of calcium transients with higher amplitude compared with that during baseline levels suggesting that the activity of these neurons may reflect the heightened homeostatic need for REMs.

## Inhibition of POA$^{GAD2}$→TMN neurons during REMs restriction reduces the REMs rebound

To investigate whether the activity of POA$^{GAD2}$→TMN neurons encodes REMs pressure and consequently facilitates the subsequent rebound in REMs, we optogenetically inhibited these neurons during the last 3 hr of REMs restriction. GAD2-Cre mice were bilaterally injected with retrograde AAVs encoding Cre-inducible SwiChR++ (AAVretro-DIO-SwiChR++-eYFP) or eYFP (AAVretro-DIO-eYFP) into the TMN followed by bilateral optic fiber implantation into the POA (*Figure 4A*). Mice underwent REMs restriction (6 hr, ZT 1.5–7.5), and laser stimulation (2 s step pulses at 60 s intervals) was applied during the last 3 hr (ZT 4.5–7.5), when the REMs pressure was highest (*Figure 4A*, *Figure 3—figure supplement 1B*). During restriction, SwiChR-mediated inhibition of POA$^{GAD2}$→TMN neurons decreased the percentage of REMs and reduced the frequency of REMs episodes with marginal significance (*Figure 4B–D*, *Figure 4—figure supplement 1A and B* amount: d=−1.254). We found that inhibition of POA$^{GAD2}$→TMN neurons during REMs restriction resulted in a reduced amount of REMs during the rebound compared with that in eYFP mice (*Figure 4B, F, and G*, *Figure 4—figure supplement 1D, E*, amount: d=−1.317). Thus, inactivating POA$^{GAD2}$→TMN neurons during heightened REMs pressure not only decreased the amount of REMs, but also prevented its homeostatic rebound during the following recovery sleep.

Finally, we examined the spectral composition of the EEG during the REMs restriction. We found that inhibition of POA$^{GAD2}$→TMN neurons during REMs restriction reduced the REMs δ power, NREMs

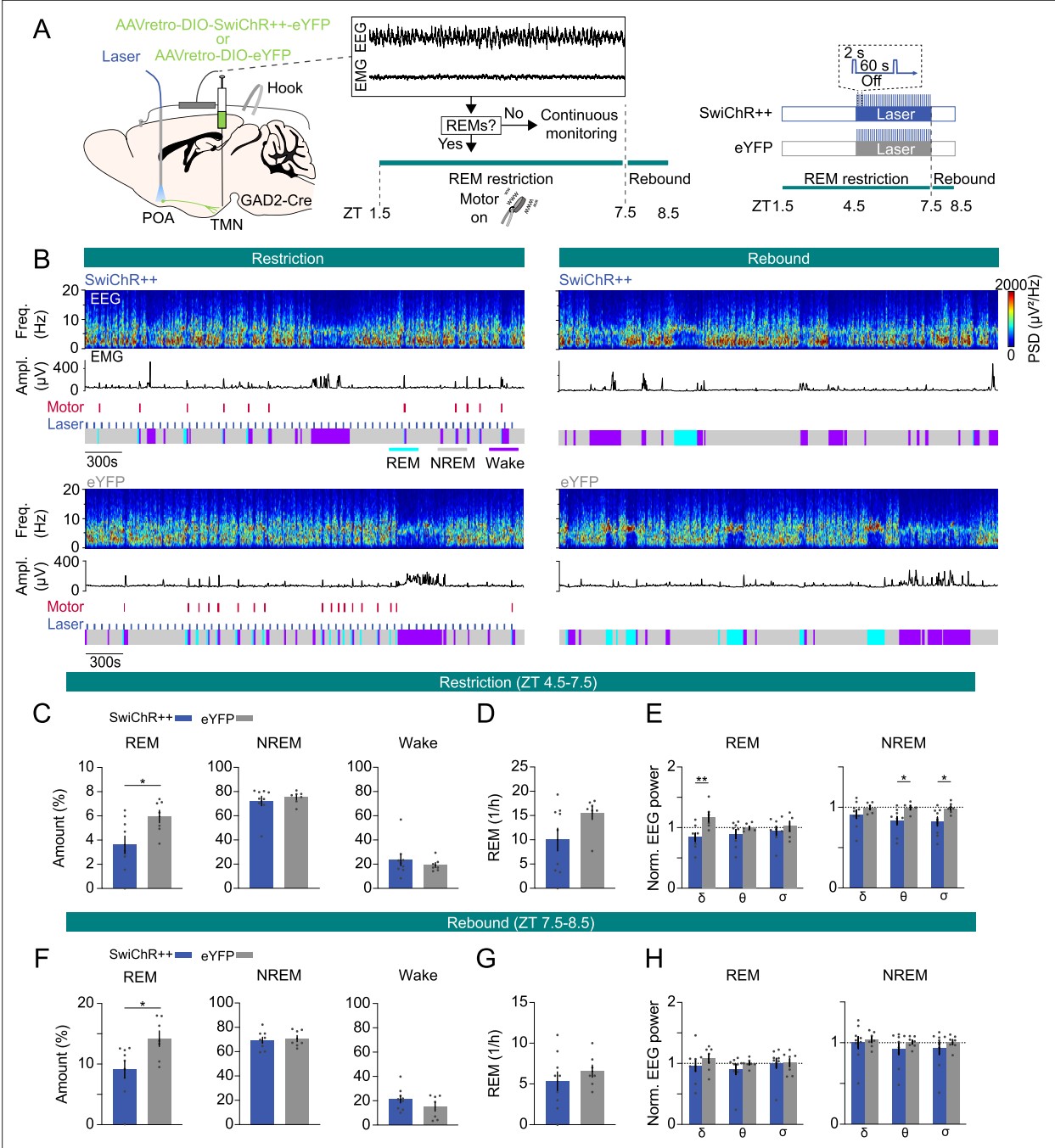

**Figure 4.** Inhibition of POA$^{GAD2}$→TMN neurons during rapid eye movement sleep (REMs) restriction attenuates the REMs rebound. (**A**) Schematic of REMs restriction/rebound and optogenetic inhibition experiments. During closed-loop REMs restriction (zeitgeber time [ZT] 1.5–7.5), a vibrating motor attached to the mouse head was used to terminate REMs. REMs was restricted for 6 hr (ZT 1.5–7.5). During the last 3 hr of restriction (ZT 4.5–7.5), laser stimulation (2 s step pulses at 60 s intervals) was applied in SwiChR++ and eYFP mice. Mouse brain figure adapted from the Allen Reference Atlas - Mouse Brain. (**B**) Example sessions from a SwiChR++ (top) and eYFP mouse (bottom) during REMs restriction with laser stimulation (left) and rebound (right). Shown are electroencephalogram (EEG) spectrogram, electromyogram (EMG) amplitude, motor vibration events, laser and color-coded brain states. (**C**) Percentage of time spent in REMs, non-rapid eye movement sleep (NREMs), and wakefulness during the last 3 hr of REMs restriction with laser stimulation (ZT 4.5–7.5) in mice expressing SwiChR++ and eYFP. Unpaired t-tests, p=0.026 for REMs amount. (**D**) Frequency of REMs episodes during the last 3 hr of REMs restriction with laser stimulation (ZT 4.5–7.5). Unpaired t-tests, p=0.0821. (**E**) Normalized EEG δ, θ, and σ power during the last 3 hr of REMs restriction with laser stimulation (ZT 4.5–7.5). Unpaired t-tests, p=0.0091, 0.0332, and 0.038 for REMs δ, NREMs θ and σ power. (**F**) Percentage of time spent in REMs, NREMs, and wakefulness during REMs rebound (ZT 7.5–8.5) in SwiChR++ and eYFP mice. Unpaired t-tests, p=0.0205

*Figure 4 continued on next page*

*Figure 4 continued*

for REMs amount. (**G**) Frequency of REMs episodes during REMs rebound (ZT 7.5–8.5). (**H**) Normalized EEG δ, θ, and σ power during REMs rebound (ZT 7.5–8.5). Bars, averages across mice; dots, individual mice; error bars, ± s.e.m. SwiChR++: n=9 mice; eYFP: n=7 mice.

The online version of this article includes the following figure supplement(s) for figure 4:

**Figure supplement 1.** Duration and frequency of rapid eye movement sleep (REMs) episodes and electroencephalogram (EEG) power during SwiChR++-mediated inhibition combined with REMs restriction and rebound.

θ and σ power compared with that in eYFP mice with laser stimulation (**Figure 4E**, **Figure 4—figure supplement 1C**). The reduced θ and σ power during NREMs could result from less attempts to enter REMs (**Figure 4D**). During rebound, there were no significant differences in EEG δ, θ, and σ power during REMs and NREMs (**Figure 4H**, **Figure 4—figure supplement 1F**).

Taken together, inhibiting POA$^{GAD2}$→TMN neurons during REMs restriction significantly decreased the amount of REMs and blocked the subsequent rebound in REMs. Our data suggest that the heightened activity of POA$^{GAD2}$→TMN neurons during sleep encodes the increased need for REMs and consequently plays an important role in the homeostatic response to REMs restriction.

## Discussion

Our study demonstrates a role of POA$^{GAD2}$→TMN neurons in the homeostatic regulation of REMs. Using fiber photometry, we showed that the POA$^{GAD2}$→TMN neurons become activated throughout NREMs before transitioning to REMs, while being most active during REMs (**Figure 1**). Sustained optogenetic inhibition of POA$^{GAD2}$→TMN neurons reduced the overall amount of REMs, and the loss of REMs was not compensated during the subsequent recovery sleep (**Figure 2**). During the period of high REMs pressure, POA$^{GAD2}$→TMN neurons exhibited an increased number of calcium transients with elevated amplitude (**Figure 3**). Optogenetic inhibition of POA$^{GAD2}$→TMN neurons during REMs restriction attenuated the subsequent rebound of REMs (**Figure 4**). Our results suggest that the activity of POA$^{GAD2}$→TMN neurons reflects an increased need for REMs in the form of enhanced calcium transients and is required for the rebound following the loss of REMs.

The TMN contains histamine-producing neurons and antagonizing histamine signaling causes sleepiness (**Watanabe et al., 1983**; **Panula et al., 1984**; **Lin et al., 1988**; **Bayliss et al., 1990**; **Haas et al., 2008**; **Uygun et al., 2016**). Consistent with our photometry recordings (**Figure 1—figure supplement 2**), electrophysiological recordings demonstrate that TMN$^{HIS}$ neurons are most active during wakefulness and less active during NREMs and REMs (**Steininger et al., 1999**; **Vanni-Mercier et al., 2003**; **John et al., 2004**; **Takahashi et al., 2006**). Throughout NREMs, the activity of TMN$^{HIS}$ neurons gradually decreased while that of POA$^{GAD2}$→TMN neurons showed an opposite pattern (**Figure 1E**, **Figure 1—figure supplement 2E**), which is likely in part the result of direct synaptic inputs from the POA$^{GAD2}$→TMN neurons to TMN$^{HIS}$ neurons (**Chung et al., 2017**; **Saito et al., 2018**). TMN$^{HIS}$ neurons in turn inhibit putative sleep-active POA neurons (**Williams et al., 2014**). Mutual inhibition between TMN$^{HIS}$ and POA$^{GAD2}$→TMN neurons may explain their antagonistic activity pattern revealed in our fiber photometry recordings.

While many studies have focused on the role of the POA in regulating NREMs (**Sherin et al., 1996**; **Szymusiak et al., 1998**; **Lu et al., 2000**; **Zhang et al., 2015**; **Kroeger et al., 2018**; **Ma et al., 2019**), previous in vivo electrophysiological studies found that the majority of sleep-active neurons in the POA are most active during REMs (**Osaka and Matsumura, 1995**; **Takahashi et al., 2009**; **Antila et al., 2022**). Similarly, recent fiber photometry recordings demonstrated that GABAergic neurons in the POA and their subtypes expressing cholecystokinin, corticotropin-releasing hormone, tachykinin 1, or galanin are most active during REMs (**Miracca et al., 2022**; **Smith et al., 2024**). Deleting the NMDA receptor GluN1 subunit in the POA reduced REMs (**Miracca et al., 2022**), and in line with this, sustained optogenetic inhibition of POA$^{GAD2}$→TMN neurons specifically decreased REMs. Consistent with these studies, our findings also support an important role of the POA in REMs regulation and, in addition, provide evidence that these neurons are also part of the homeostat regulating REMs.

A previous study showed that the number of c-Fos positive POA neurons is positively correlated with the amount of REMs in rats (**Lu et al., 2002**). Moreover, REMs restriction led to an increase in the number of POA neurons expressing c-Fos, which was correlated with the number of attempts to enter REMs (**Gvilia et al., 2006**). Together with our findings that the number of calcium

transients of POA$^{GAD2}$→TMN neurons increased during REMs restriction and that inhibition of these neurons blocked the following REMs rebound, these results support a crucial role of the POA in the homeostatic regulation of REMs. For the future, it would be interesting to test whether POA neurons projecting to other postsynaptic areas are also involved in the homeostatic regulation of REMs. A previous study showed that POA neurons projecting to REMs-regulatory pontine regions including the laterodorsal tegmental nucleus, locus coeruleus (LC), and dorsal raphe nucleus express increased levels of c-Fos after periods of dark exposure that increased REMs (*Lu et al., 2002*). However, the number of c-Fos positive POA neurons projecting to the LC was not increased upon REMs restriction, suggesting that this subpopulation may not be involved in the homeostatic regulation of REMs (*Verret et al., 2006*). Besides the TMN, the POA also projects to other REMs-regulatory regions such as the ventrolateral periaqueductal gray (vlPAG) and lateral hypothalamus (*Steininger et al., 2001*; *Saito et al., 2018*). Particularly, the projections to the vlPAG are of interest for future research, as GABAergic neurons in this area have been previously implicated in the homeostatic regulation of REMs (*Hayashi et al., 2015*; *Weber et al., 2018*). It remains to be tested whether POA$^{GAD2}$→TMN neurons also project to these brain regions to potentially regulate REMs homeostasis.

The cellular mechanisms underlying the elevated activity of POA neurons during high REMs pressure are unknown. Sleep-promoting neurons in the dorsal fan-shaped body of *Drosophila* display increased intrinsic neuronal excitability in response to sleep need (*Donlea et al., 2014*). REMs deprivation was shown to change the intrinsic excitability of hippocampal neurons and impact synaptic plasticity (*McDermott et al., 2003*; *Mallick and Singh, 2011*; *Zhou et al., 2020*). The elevated activity of POA$^{GAD2}$→TMN neurons during heightened REMs pressure may similarly be the result of an increased excitability. Given that the POA is also involved in the homeostatic regulation of NREMs (*Sherin et al., 1996*; *Szymusiak et al., 1998*; *Gong et al., 2004*; *Zhang et al., 2015*; *Ma et al., 2019*; *Lu et al., 2000*), it would be interesting to study how different POA subpopulations integrate the homeostatic need for NREMs and REMs.

Together, we have demonstrated a role of POA$^{GAD2}$→TMN neurons in the homeostatic regulation of REMs. REMs disturbances are observed in a variety of psychiatric disorders such as depression and PTSD and often precede their clinical onset (*Ross et al., 1989*; *Gottesmann and Gottesman, 2007*; *Germain, 2013*; *Palagini et al., 2013*). Elucidating the circuit mechanisms underlying the homeostatic regulation of REMs may provide novel therapeutic targets to specifically regulate and normalize REMs in these psychiatric disorders to alleviate associated symptoms and potentially slow down their progression.

## Materials and methods

### Mice

All experimental procedures were approved by the Institutional Animal Care and Use Committee (IACUC reference # 806197) at the University of Pennsylvania and conducted in compliance with the National Institutes of Health Office of Laboratory Animal Welfare Policy. Experiments were performed in male and female GAD2-IRES-Cre mice (#010802, Jackson Laboratory, generously donated by *Taniguchi et al., 2011*) or HDC-IRES-Cre mice (#021198, Jackson Laboratory, generously donated by *Zecharia et al., 2012*) aged 10–18 weeks, weighing 18–25 g at the time of surgery. Animals were group-housed with littermates on a 12 hr light/12 hr dark cycle (lights on 7 am and off 7 pm) with ad libitum access to food and water.

### Viruses

Cre-dependent adeno-associated viral vectors were used to selectively express GCaMP, SwiChR++, or eYFP in POA$^{GAD2}$→TMN neurons, POA$^{GAD2}$→TMN projections, or TMN$^{HIS}$ neurons. pGP-AAV-Syn-FLEX-jGCaMP8s-WPRE was developed from the GENIE Project (*Zhang et al., 2023*) (162377-AAVrg, Addgene). pAAV-Syn-FLEX-GCaMP6s-WPRE-SV40 was developed from Douglas Kim & GENIE Project (*Chen et al., 2013*) (Penn Vector Core or 100845-AAV1, Addgene).

rAAV$_2$-Retro-Ef1α-DIO-SwiChR++-eYFP (R47730, UNC Vector Core).

rAAV$_2$-Retro-Ef1α-DIO-eYFP (R49556, UNC Vector Core).

## Surgical procedures

All procedures followed the IACUC guidelines for rodent survival surgery. Mice were anesthetized with isoflurane (1–2%) during the surgery, and placed on a stereotaxic frame (Kopf) while being on a heating pad to maintain body temperature. The skin was incised and small holes were drilled for virus injections and implantations of optic fibers and EEG/EMG electrodes.

For fiber photometry experiments to image POA$^{GAD2}$→TMN neurons (*Figures 1 and 3*), pGP-AAV-Syn-FLEX-jGCaMP8s-WPRE was injected (Nanoject II, Drummond Scientific) into the TMN (300 nl; AP –2.4 mm; ML –1 mm; DV –5.4 to –5.2 mm, relative to bregma) and an optic fiber (400 μm diameter) was implanted into the POA (AP 0.2 mm; ML –0.6 mm; DV –5.2 mm). For imaging the POA$^{GAD2}$→TMN axonal fibers (*Figure 1—figure supplement 1*), pAAV-Syn-FLEX-GCaMP6s-WPRE-SV40 was injected into the POA (300 nl) and an optic fiber was implanted into the TMN. To image TMN$^{HIS}$ neurons (*Figure 1—figure supplement 2*), pAAV-Syn-FLEX-GCaMP6s-WPRE-SV40 was injected into the TMN (300 nl) and an optic fiber was implanted into the TMN.

For optogenetic inhibition experiments (*Figures 2 and 4*), rAAV$_2$-Retro-Ef1α-DIO-SwiChR++-eYFP (for inhibition group) or rAAV$_2$-Retro-Ef1α-DIO-eYFP (for control group) was bilaterally injected into the TMN (300 nl) and bilateral optic fibers (200 μm diameter) were implanted into the POA (AP 0.2 mm; ML ±1.5 mm [angled at 10°]; DV –5.2 mm).

All mice were implanted with electroencephalogram (EEG) and electromyogram (EMG) electrodes. EEG signals were recorded with stainless steel wires attached to two screws, located in the skull on top of the parietal (AP –2 mm; ML 2 mm) and frontal cortex (AP 1.7 mm; ML 0.6 mm). A reference screw was inserted on top of the cerebellum. Two EMG electrodes were inserted into the neck musculature. The incision was closed with suture and the EEG/EMG electrodes and optic fibers were secured to the skull using dental cement (A-M Systems). We performed optogenetic and fiber photometry experiments at least 4 weeks after surgery.

## Immunohistochemistry

Mice were deeply anesthetized and transcardially perfused with phosphate-buffered saline (PBS) followed by 4% paraformaldehyde (PFA) in PBS. Brains were removed and fixed overnight in 4% PFA in PBS and then stored in 30% sucrose in PBS. Brains were embedded with OCT compound (Tissue-Tek, Sakura Finetek) and frozen. 40 μm sections were cut using a cryostat (Thermo Scientific HM525 NX) and directly mounted onto glass slides. Brain sections were washed in PBS for 5 min, permeabilized using PBST (0.3% Triton X-100 in PBS) for 30 min, and incubated in blocking solution (5% normal donkey serum in 0.3% PBST; 017-000-001, Jackson ImmunoResearch Laboratories) for 1 hr. Brain sections were incubated with chicken anti-GFP antibody (1:1000; GFP8794984, Aves Lab) in the blocking solution overnight at 4°C. The following morning, sections were washed in PBS and incubated for 3 hr with the donkey anti-chicken secondary antibody conjugated to a green Alexa fluorophore 488 (1:500; 703-545-155, Jackson ImmunoResearch Laboratories). Afterward, sections were washed with PBS followed by counterstaining with Hoechst solution (#33342, Thermo Scientific). Slides were coverslipped with mounting medium (Fluoromount-G, Southern Biotechnic) and imaged using a fluorescence microscope (Microscope, Leica DM6B; Camera, Leica DFC7000GT; LED, Leica CTR6 LED) to verify virus expression and optic fiber placement. Animals were excluded if no virus expression is detected or the virus expression/optic fiber tips were not properly localized to the targeted area.

## Viral transfection mapping

We generated heatmaps of the virus expression across mice as previously described (*Smith et al., 2024*; *Stucynski et al., 2022*; *Schott et al., 2023*). Coronal reference images for the corresponding AP coordinates were downloaded from the Allen Reference Atlas - Mouse Brain (atlas.brain-map.org). For a given AP reference atlas section, the corresponding histology section from each mouse was overlaid and regions in which GCaMP labeled cell bodies were present were manually outlined. Custom Python programs detected these outlines and determined for each location on the reference picture the number of mice with overlapping virus expression, which was encoded using different green color intensities.

## Polysomnographic recordings

All sleep recordings were performed in a cage to which the animal had been habituated for several days. All recordings were performed during the light phase between 8 am and 5 pm (ZT1–10) in

sound-attenuating chambers. For sleep recordings, EEG and EMG signals were recorded using an RHD2132 amplifier (Intan Technologies, sampling rate 1 kHz) connected to an RHD USB interface board (Intan Technologies). For fiber photometry experiments, a calcium signal was recorded using an RZ5P amplifier (Tucker-Davis Technologies, sampling rate 1.5 kHz). EEG and EMG signals were referenced to a ground screw located on top of the cerebellum. At the start of each sleep recording, EEG and EMG electrodes were connected to flexible recording cables via small connectors. To determine the brain state of the animal, we first computed the EEG and EMG spectrogram for sliding, half-overlapping 5 s windows, resulting in 2.5 s time resolution. To estimate within each 5 s window the power spectral density (PSD), we performed Welch's method with Hanning window using sliding, half-overlapping 2 s intervals. Next, we computed the time-dependent δ (0.5–4 Hz), θ (5–12 Hz), σ (12–20 Hz), and high γ (100–150 Hz) power by integrating the EEG power in the corresponding ranges within the EEG spectrogram. In addition, we calculated the ratio of the θ and δ power (θ/δ) and the EMG power in the range of 50–500 Hz. For each power band, we used its temporal mean to separate it into a low and high part (except for the EMG and θ/δ ratio, where we used the mean plus one standard deviation as threshold). REMs was defined by a high θ/δ ratio, low EMG, and low δ power. NREMs was defined by high δ power, a low θ/δ ratio, and low EMG power. In addition, states with low EMG power, low δ power, but high σ power were scored as NREMs. Wake was defined by low δ power, high EMG power, and high γ power (if not otherwise classified as REMs). Our automatic algorithm that has been previously used in *Weber et al., 2015*; *Weber et al., 2018*; *Chung et al., 2017*; *Antila et al., 2022*; *Smith et al., 2024*; *Stucynski et al., 2022*; *Schott et al., 2023*, has 90.256% accuracy compared with the manual scoring by expert annotators. We manually verified the automatic classification using a graphical user interface visualizing the raw EEG and EMG signals, EEG spectrograms, EMG amplitudes, and the hypnogram to correct for errors, by visiting each single 2.5 s epoch in the hypnograms. The software for automatic brain state classification and manual scoring was programmed in Python (https://github.com/tortugar/Lab/tree/tortugar-patch-1/PySleep copy archived at *tortugar, 2024*).

## Fiber photometry

Prior to the recording, the optic fiber and EEG/EMG electrodes were connected to flexible patch cables. For calcium imaging, a first LED (Doric lenses) generated the excitation wavelength of 465 nm and a second LED emitted 405 nm light, which served as control for bleaching and motion artifacts. 465 and 405 nm signals were modulated at two different frequencies, 210 and 330 Hz respectively. Both lights traveled through dichroic mirrors (Doric lenses) before entering a patch cable attached to the optic fiber. Fluorescence signals emitted by GCaMP8s or GCaMP6s were collected by the optic fiber and traveled via the patch cable through a dichroic mirror and GFP emission filter (Doric lenses) before entering a photoreceiver (Newport Co.). Photoreceiver signals were relayed to an RZ5P amplifier and demodulated into two signals using the Synapse software (Tucker-Davis Technologies), corresponding to the 465 and 405 nm excitation wavelengths. To analyze the calcium activity, we used custom-written Python scripts. First, both signals were low-pass filtered at 2 Hz using a fourth order digital Butterworth filter. Next, we fitted the 405 nm to the 465 nm signal using linear regression. Finally, the linear fit was subtracted from the 465 nm signal to correct for photobleaching and/or motion artifacts, and the difference was divided by the linear fit yielding the ΔF/F signal. Both the fluorescence signals and EEG/EMG signals were simultaneously recorded using the RZ5P amplifier. Fiber photometry recordings were excluded if the signals suddenly shifted, likely due to a loose connection between the optic fiber and patch cable.

## Optogenetic manipulation

Sleep recordings were performed during the light phase (ZT2–8). Mice were tethered to bilateral patch cables connected with the lasers and a flexible recording cable to record EEG/EMG signals. The recording started after 30 min of habituation. For optogenetic inhibition experiments, 2 s step pulses (1–3 mW, 60 s intervals) were generated by a blue laser (473 nm, Laserglow) and sent through the optic fiber (200 μm diameter, Thorlabs) connected to the ferrule on the animal's head for 3 hr (ZT2–5). This laser stimulation protocol was rationally designed based on previous reports of sustained inhibition and prior results that recapitulate similar findings as inhibitory chemogenetic techniques (*Iyer et al., 2016*; *Kim et al., 2016*; *Wiegert et al., 2017*; *Stucynski et al., 2022*).

TTL pulses to trigger the laser were controlled using a Raspberry Pi, which was controlled by a custom-programmed user interface programmed in Python. Following sustained inhibition, an additional 3 hr (ZT5–8) recording was performed without laser stimulation (post-laser session). Baseline recordings were performed (without laser stimulation, ZT2–5), and counterbalanced across mice and days to avoid potential order effects. For each mouse, we collected two to three baseline and laser recordings each.

## REMs restriction

We employed an automatic REMs detection algorithm (*Weber et al., 2015*; *Weber et al., 2018*; *Stucynski et al., 2022*; *Schott et al., 2023*) and used a small vibrating motor to terminate/restrict REMs (*Cardis et al., 2021*; *Osorio-Forero et al., 2023*; https://github.com/luthilab/IntanLuthiLab, copy archived at *luthilabcol, 2023*). A small vibrating motor (DC 3 V Mini Vibration Motor, diameter: 10 mm, thickness: 3 mm, BOJACK) with a soldered lobster claw clasp was attached to a small wire hook secured in the dental cement of each mouse head. The motors had cables that were connected to a Raspberry Pi which controlled the motor onset and offset.

The automatic REMs detection algorithm determined whether the mouse was in REMs or not based on real-time spectral analysis of the EEG/EMG signals. The onset of REMs was defined as the time point where the EEG θ/δ ratio exceeded a threshold (mean + 1 std of θ/δ), which was calculated from the same mouse using previous recordings. As soon as a REMs episode was detected, the small vibrating motor turned on to terminate REMs and consequently woke up the mouse. The motor vibrated until the REMs episode was terminated, i.e., when the θ/δ ratio dropped below its mean value or if the EMG amplitude surpassed a threshold (mean + 0.5 std of amplitude). All REMs restriction experiments started at ZT1.5 and lasted until ZT7.5. Following the restriction, motors were turned off and mice were permitted to enter recovery sleep.

To monitor the calcium activity during REMs restriction using fiber photometry, we performed manual REMs restriction to avoid potential motion artifacts caused by the vibrating motor that could interfere with the integrity of the fiber photometry signals. Briefly mice underwent a 4 hr automatic REMs restriction protocol as described above. During the last 2 hr of restriction, REMs was detected manually based on the EEG and EMG signal by an experimenter who gently pulled on a string attached to the hook secured in the dental cement of the mouse head. To avoid potential photobleaching, the recording was performed during the last 1 hr of REMs restriction and 1 hr of REMs rebound. Each mouse also underwent a baseline recording on a separate day. The order of REMs restriction and baseline recordings were varied to minimize the impact of the experimental sequence on the results.

We also performed automatic REMs restriction combined with optogenetic inhibition. Each mouse underwent REMs restriction for 6 hr, and we continuously delivered 2 s step pulses (1–3 mW, 473 nm, Laserglow) at 60 s intervals during the last 3 hr of restriction (ZT4.5–7.5). Following REMs restriction, the recovery sleep was recorded.

## Analysis of ΔF/F activity at brain state transitions and during NREMs

To calculate the neural activity changes relative to brain state transitions (*Figure 1D*, *Figure 1—figure supplements 1D and 2D*), we aligned the ΔF/F signals for transitions across all mice relative to the time point of the transition (t=0 s). For each NREMs→ REMs or NREMs→wake transition, we ensured that the preceding NREMs episodes lasted for at least 60 s, only interrupted by short awakenings (≤20 s). To determine the time point at which the activity significantly started to increase or decrease, we used the first 10 s of NREMs as baseline. For REMs→wake and wake→NREMs transitions, the preceding REMs or wake episode was at least 30 s long. Using one-way repeated measures (rm) ANOVA, we tested whether the activity (downsampled to 10 s bins) within each 10 s bin was significantly modulated throughout the transition (from –60 to 30 s). Finally, using pairwise t-tests with Holm-Bonferroni correction, we determined the time bins for which the activity significantly differed from the baseline bin (activity for bin –60 to –50 s). The time point for a given 10 s bin was set to its midpoint. To account for multiple comparisons, we divided the significance level (α=0.05) by the number of comparisons (Bonferroni correction). To analyze the activity throughout NREMs, we normalized the duration of all NREMs episodes and the corresponding ΔF/F signals to the same length (*Figure 1E*, *Figure 1—figure supplement 2D and E*).

## Spectrotemporal correlation analysis

To identify features of the EEG spectrogram associated with POA$^{GAD2}$→TMN calcium activity, we adapted a receptive field model (**Weber et al., 2010**; **Schott et al., 2023**) to predict POA$^{GAD2}$→TMN neural activity from the spectrogram. Intuitively, we estimated a spectrotemporal filter using linear regression that predicts for each time point the POA$^{GAD2}$→TMN calcium response. In more detail, we first computed the EEG spectrogram $E(f_i, t_j)$ using 2 s windows with 80% overlap, resulting in a time resolution dt of 400 ms. Each spectrogram frequency was normalized by its mean power across the recording, and the parameter $E(f_i, t_j)$ specifies the relative amplitude of frequency $f_i$ for time point $t_j$. We then downsampled the ΔF/F response using the same time resolution as for the spectrogram, and extracted all time bins with REMs, NREMs, or wake for analysis.

To predict the calcium response, the EEG spectrogram was linearly filtered with the kernel $H(f_i, t_l)$. In analogy to receptive fields estimated using similar approaches for sensory neurons, we used the term 'spectral field' for $H(f_i, t_l)$. The spectral field can be described as the set of coefficients that optimally relate the neural activity at time $t_j$ to the EEG spectrogram at time $t_{j+l}$, where l represents the lag between the time points of the spectrogram and the neural response. The estimated frequency components of $H(f_i, t_l)$ ranged from $f_1 = 0.5$ Hz to $f_{nf} = 20$ Hz, while the time axis ranged from $-n_T * dt = -50$ s to $n_T * dt = 50$ s. Thus, $H(f_i, t_l)$ comprises $n_f$ frequencies and $2 * n_T + 1$ time bins over a window from −50 to +50 s relative to the neural response. The convolution of the EEG spectrogram with the spectral field can be expressed as

$$\hat{r}(t_j) = r_0 + \sum_{l=-n_r}^{n_r} \sum_{i=1}^{n_f} E(f_i, t_{j+l}) \, H(f_i t_i) \tag{1}$$

The scalar parameter $r_0$ denotes a constant offset, and $r(t|l|j)$ denotes the predicted neural response. The optimal spectral field $H(f_i, t_l)$ minimizes the mean-squared error between the predicted and measured neural response at time $t_j$. To account for the large number of estimated parameters, we included a regularization term in the error function, which penalizes large kernel components and therefore guards against overfitting of the model. The spectral field for each recording was estimated using fivefold cross-validation to determine the regularization parameter ($\lambda$) that optimized the average model performance on the test sets. Kernels were averaged across all recordings for individual animals; the spectrotemporal correlation in **Figure 1F** represents the mean spectral field across all mice.

## PSD and power estimation

The PSD of the EEG was computed using Welch's method with the Hanning window for sliding, half-overlapping 2 s intervals. To calculate the power within a given frequency band, we approximated the corresponding area under the spectral density curve using a midpoint Riemann sum. To compute the EMG amplitude, we calculated the PSD of the EMG and integrated frequencies between 5 and 100 Hz. To test whether laser stimulation changed the spectral density during a specific brain state, we determined for each mouse the δ, θ, or σ power for that state with and without laser. To compare the PSD between experimental and control mice, we calculated the δ, θ, and σ power with and without laser stimulation for each mouse and normalized the power values with laser by the power obtained for epochs without laser.

## Detection of calcium transients

To detect calcium transients, we first filtered the ΔF/F signal with a zero-lag, fourth order digital Butterworth filter with cutoff frequency of 1/20 Hz. Next, prominent peaks in the signal were detected using the function scipy.find_peaks provided by the open source Python library scipy (https://scipy.org). As parameter for the peak prominence, we used 0.15 * distance between the 1st and 99th percentile of the distribution of the ΔF/F signal. A transient was defined as occurring during NREMs and REMs, if the peak overlapped with NREMs or REMs respectively. The calcium transient amplitude was calculated by using the values 10 s preceding the peak and subtracting that from the peak values at 0 s.

## Statistical tests

Statistical analyses were performed using the Python modules (scipy.stats, https://scipy.org; pingouin, https://pingouin-stats.org) and Prism v9.5.0.0 (GraphPad Software Inc). We did not predetermine sample sizes, but cohorts were similarly sized as in other relevant sleep studies (*Jego et al., 2013*; *Ma et al., 2019*). All data collection was randomized and counterbalanced. All data are reported as mean ± s.e.m. A (corrected) p-value <0.05 was considered statistically significant for all comparisons. Data were compared using unpaired t-tests, paired t-tests, one-way ANOVAs, or two-way ANOVAs followed by multiple comparisons as appropriate. The statistical results for the figures are presented in the *Supplementary file 1* and Figure legends.

## Data and code sharing plans

The code used for data analysis is publicly available under: https://github.com/tortugar/Lab, copy archived at *tortugar, 2024*. All the data have been deposited at Zenodo (https://zenodo.org) and are available as of the date of publication.

## Acknowledgements

We thank Mandy Schott for generating Python scripts for optogenetic manipulation during REMs restriction, Jenny Smith for help with generating the virus expression heatmaps and members of the Chung and Weber labs for helpful discussion. This work was funded by the National Institute of Neurological Disorders and Stroke (R01-NS-110865), the Whitehall Foundation, the Alfred P Sloan Foundation, a NARSAD Young Investigator Grant from the Brain & Behavior Research Foundation, a Simons Foundation Pilot Award, an Eagle Autism Challenge Pilot Grant, the Thomas B and Jeannette E Laws McCabe Fund Award, the Hartwell Individual Biomedical Research Award (to SC), and the NIH individual F31 fellowship (NS118963-01A1, to JM).

## Additional information

### Funding

| Funder | Grant reference number | Author |
|---|---|---|
| National Institute of Neurological Disorders and Stroke | NS110865 | Shinjae Chung |
| Whitehall Foundation | | Shinjae Chung |
| Alfred P. Sloan Foundation | | Shinjae Chung |
| Brain and Behavior Research Foundation | | Shinjae Chung |
| Simons Foundation Autism Research Initiative | | Shinjae Chung |
| Eagles Autism Foundation | | Shinjae Chung |
| Hartwell Foundation | | Shinjae Chung |
| National Institute of Neurological Disorders and Stroke | NS118963-01A1 | John J Maurer |

The funders had no role in study design, data collection and interpretation, or the decision to submit the work for publication.

### Author contributions

John J Maurer, Conceptualization, Resources, Data curation, Software, Formal analysis, Funding acquisition, Validation, Investigation, Visualization, Methodology, Writing - original draft, Writing - review and editing; Alexandra Lin, Data curation, Validation, Investigation; Xi Jin, Software, Methodology; Jiso Hong, Methodology; Nicholas Sathi, Validation; Romain Cardis, Alejandro Osorio-Forero,

Anita Lüthi, Methodology, Writing - review and editing; Franz Weber, Conceptualization, Resources, Software, Methodology, Writing - review and editing; Shinjae Chung, Conceptualization, Resources, Supervision, Funding acquisition, Methodology, Writing - review and editing

**Author ORCIDs**
John J Maurer ⓘ http://orcid.org/0000-0003-2710-1963
Alejandro Osorio-Forero ⓘ http://orcid.org/0000-0003-4341-4206
Anita Lüthi ⓘ http://orcid.org/0000-0002-4954-4143
Shinjae Chung ⓘ http://orcid.org/0000-0003-2268-563X

**Ethics**
All experimental procedures were approved by the Institutional Animal Care and Use Committee (IACUC reference # 806197) at the University of Pennsylvania and conducted in compliance with the National Institutes of Health Office of Laboratory Animal Welfare Policy.

Reviewer #1 (Public Review): https://doi.org/10.7554/eLife.92095.3.sa1
Reviewer #2 (Public Review): https://doi.org/10.7554/eLife.92095.3.sa2
Author response https://doi.org/10.7554/eLife.92095.3.sa3

---

## Additional files

### Supplementary files
• MDAR checklist
• Supplementary file 1. Statistical Analysis Table.

### Data availability
All data have been deposited at Zenodo at https://doi.org/10.5281/zenodo.11107281.

The following dataset was generated:

| Author(s) | Year | Dataset title | Dataset URL | Database and Identifier |
|---|---|---|---|---|
| Maurer J | 2024 | Homeostatic regulation of REM sleep by the preoptic area of the hypothalamus | https://doi.org/10.5281/zenodo.11107281 | Zenodo, 10.5281/zenodo.11107281 |

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
