## [Editor Report · eLife assessment]

This **valuable** study advances our understanding of the brain nuclei involved in rapid-eye movement (REM) sleep regulation. Using a combination of imaging, electrophysiology, and optogenetic tools, the study provides **convincing** evidence that inhibitory neurons in the preoptic area of the hypothalamus influence REM sleep. This work will be of interest to neurobiologists working on the brain circuits of sleep.

---

## [Referee Report · Reviewer #1 (Public Review)]

This paper identifies GABA cells in the preoptic hypothalamus and others in the posterior hypothalamus which are involved in REM sleep rebound (the increase in REM sleep) after selective REM sleep deprivation. By calcium photometry, these preoptic cells are most active during REM, and show more calcium signals during REM deprivation, suggesting they respond to "REM pressure". Inhibiting these cells ontogenetically diminishes REM sleep. The optogenetic and photometry work is carried out to a high standard, the paper is well written, and the findings are interesting and enhance our understanding of REM sleep regulation. The new findings make it clear that as for the circuitry that regulates NREM sleep, REM sleep circuitry is also quite distributed in the brain. It is unclear if there is a true "REM center". The study of mechanisms of catching up on lost sleep (sleep homeostasis), has previously focused on NREM sleep, where various circuits have been identified. That there is a special mechanism that also tracks time awake and compensates with REM sleep is intriguing.

In a broader context, the existence of REM rebound suggests that REM sleep must have a function, otherwise why catch up on it. There is a lot of literature that suggests REM contributes to emotional processing, for example. The new findings deepen our appreciation of REM regulation. As REM sleep is often disturbed in stress (e.g. post-traumatic stress disorder) and in depression, understanding more about REM regulation could ultimately aid treatments for people living with these conditions.

---

## [Referee Report · Reviewer #2 (Public Review)]

Maurer et al investigated the contribution of GAD2+ neurons in the preoptic area (POA), projecting to the tuberomammillary nucleus (TMN), to REM sleep regulation. They applied an elegant design to monitor and manipulate activity of this specific group of neurons: a GAD2-Cre mouse, injected with retrograde AAV constructs in the TMN, thereby presumably only targeting GAD2+ cells projecting to the TMN. Using this set-up in combination with technically challenging techniques including EEG with photometry and REM sleep deprivation, the authors found that this cell-type studied becomes active shortly (≈40sec) prior to entering REM sleep and remains active during REM sleep. Moreover, optogenetic inhibition of GAD2+ cells inhibits REM sleep by a third, and also impairs the rebound in REM sleep in the following hour. Thus, the data makes a convincing case for a role of GAD2+ neurons in the POA projecting to the TMN in REM sleep regulation.

---

## [Author Response]

The following is the authors’ response to the original reviews.

**eLife assessment**
This valuable study advances our understanding of the brain nuclei involved in rapid-eye movement (REM) sleep regulation. Using a combination of imaging, electrophysiology, and optogenetic tools, the study provides convincing evidence that inhibitory neurons in the preoptic area of the hypothalamus influence REM sleep. This work will be of interest to neurobiologists working on sleep and/or brain circuitry.
**Public Reviews:**

**Reviewer #1 (Public Review):**
Summary:This paper identifies GABA cells in the preoptic hypothalamus which are involved in REM sleep rebound (the increase in REM sleep) after selective REM sleep deprivation. By calcium photometry, these cells are most active during REM, and show more claim signals during REM deprivation, suggesting they respond to "REM pressure". Inhibiting these cells ontogenetically diminishes REM sleep. The optogenetic and photometry work is carried out to a high standard, the paper is well-written, and the findings are interesting.

We thank the reviewer for the detailed feedback and thoughtful comments on how to improve our manuscript. To address the reviewer’s concerns, we revised our discussion and added new data. Below, we address the concerns point by point.

Points that could be addressed or discussed:(1) The circuit mechanism for REM rebound is not defined. How do the authors see REM rebound as working from the POAGAD2 cells? Although the POAGAD2 does project to the TMN, the actual REM rebound could be mediated by a projection of these cells elsewhere. This could be discussed.

We demonstrate thatPOA GAD2→TMN cells become more frequently activated as the pressure for REMs builds up, whereas inhibiting these neurons during high REMs pressure leads to a suppression of the REMs rebound. It is not known how POA GAD2→TMN cells encodeincreased REMs pressure and subsequently influence the REMs rebound. REMsdeprivation wasshown to changethe intrinsic excitabilityof hippocampal neurons and impact synaptic plasticity (McDermott et al., 2003; Mallick and Singh, 2011 ; Zhou et al., 2020) . We speculate that increasedREMs pressure leads to an increase in the excitabilityof POA->TMN neurons, reflected inthe increased number ofcalcium peaks. The increased excitability of POA GAD2→TMN neurons in turn likely leads to stronger inhibition of downstream REM-off neurons. Consequently, as soon as REMsdeprivation stops, there is an increased chance for enteringREMs. The time coursefor how long it takes till the POA excitability resettles toits baseline consequently sets a permissive time window for increasedamounts of REMs to recover its lostamount. For future studies, it would be interesting to map how quickly the excitability ofPOA neurons increases or decays as afunction of the lost or recovered amount of REMs andunravel the cellularmechanisms underlying the elevated activity of POAGAD2 →TMN neurons during highREMs pressure, e.g., whether changes in the expression of ion channels contribute to increasedexcitability of these neurons (Donlea et al., 2014) . As we mentioned in the Discussion, the POAalso projects to other REMs regulatorybrain regions such as the vlPAG and LH. Therefore, it remains to be tested whether POA GAD2 →TMN neurons also innervate these brain regions to potentially regulate REMs homeostasis. We explicitly state this now in the revised Discussion.

(2) The "POAGAD2 to TMN" name for these cells is somewhat confusing. The authors chose this name because they approach the POAGAD2 cells via retrograde AAV labelling (rAAV injected into the TMN). However, the name also seems to imply that neurons (perhaps histamine neurons) in the TMN are involved in the REM rebound, but there is no evidence in the paper that this is the case. Although it is nice to see from the photometry studies that the histamine cells are selectively more active (as expected) in NREM sleep (Fig. S2), I could not logically see how this was a relevant finding to REM rebound or the subject of the paper. There are many other types of cells in the TMN area, not just histamine cells, so are the authors suggesting that these non-histamine cells in the TMN could be involved?

We acknowledge that other types of neurons in the TMN may also be involved in the REMs rebound, and therefore inhibition of histamine neurons by POA GAD2 →TMN neurons may not be the sole source of the observed effect. To stress that other neurons within the TMN and/or brain regions may also contribute to the REMs rebound, we have revised the Results section.

We performed complementary optogenetic inhibition experiments of TMN HIS neurons to investigate if suppression of these neurons is sufficient to promote REMs. We foundthat SwiChR++ mediated inhibition of TMNHIS neurons increased theamount of REMs compared withrecordings without laser stimulation in the same mice and eYFPmice withlaser stimulation. Thus, while TMN HIS neurons may not bethe only downstream target of GABAergic POA neurons, these data suggest that they contribute to REMs regulation. We have incorporated these results in Fig. S4 .

We further investigated whether the activity of TMN HIS neurons changes between two REMs episodes. Assumingthat REMs pressure inhibits the activity ofREM-off histamine neurons,their firing rates should behighest right after REMs ends when REMs pressure is lowest, and progressivelydecay throughout the inter-REM interval, and reach their lowest activity right before the onset of REMs ( Park et al., 2021) , similarto the activity profile observed for vlPAG REM-off neurons (Weber et al., 2018).We indeed found that TMNHIS neurons displaya gradual decrease in their activity throughout theinter-REM interval and thus potentially reflect the build up of REM pressure ( Fig. S2F ).

(3) It is a puzzle why most of the neurons in the POA seem to have their highest activity in REM, as also found by Miracca et al 2022, yet presumably some of these cells are going to be involved in NREM sleep as well. Could the same POAGAD2-TMN cells identified by the authors also be involved in inducing NREM sleep-inhibiting histamine neurons (Chung et al). And some of these POA cells will also be involved in NREM sleep homeostasis (e.g. Ma et al Curr Biol)? Is NREM sleep rebound necessary before getting REM sleep rebound? Indeed, can these two things (NREM and REM sleep rebound) be separated?

Previous studies have demonstrated that POA GABAergic neurons, including those projecting to the TMN, are involved in NREMs homeostasis (Sherin et al., 1998; Gong et al., 2004; Ma et al., 2019) . Therefore, we predict that POA neurons that are involved in NREMs homeostasis are a subset of POA GAD2 → TMN neurons in our manuscript.

Using optrode recordings in the POA, we recently reported that 12.4% of neurons sampled have higher activity during NREMs compared with REMs; in contrast, 43.8% of neurons sampled have the highest activity during REMs compared with NREMs (Antila et al., 2022) indicating that the proportion of NREM max neurons is smaller compared with REM max neurons. These proportions of neurons are in agreement with previous results (Takahashi et al., 2009) . Considering fiber photometry monitors the average activity of a population of neurons as opposed to individual neurons, it is possible that we recorded neural activity across heterogeneous populations and therefore our findings may disguise the neural activity of the low proportion of NREMs neurons. We previously reported thespiking activity of POA GAD2 →TMN neurons at the singlecell level (Chung et al., 2017) . We have noted in themanuscript thatwhile the activity ofPOA GAD2→TMN neurons is highestduring REMs, theneural activity increases at NREMs → REMs transitions indicating these neurons also areactive during NREMs.

Using our REMs restriction protocol, we selectively restricted REMs leading to the subsequent rebound of REMs without affecting NREMs and consequently we did not find an increase in the amount of NREMs during the rebound or an increase in slow-wave activity, a key characteristic of sleep rebound that gradually dissipates during recovery sleep (Blake and Gerard, 1937; Williams et al., 1964; Rosa and Bonnet, 1985; Dijk et al., 1990; Neckelmann and Ursin, 1993; Ferrara et al., 1999) . However, during total sleep deprivation when subjects are deprived of both NREMs and REMs, isolating NREMs and REMs rebound may not be attainable.

(4) Is it possible to narrow down the POA area where the GAD2 cells are located more precisely?

POA can be subdivided into anatomically distinct regions such as medial preoptic area, median preoptic area, ventrolateral preoptic area, and lateral preoptic area (MPO, MPN, VLPO, and LPO respectively). To quantify where the virus expressing GAD2 cells and optic fibers are located within the POA, we overlaid the POA coronal reference images (with red boundaries denoting these anatomically distinct regions) over the virus heat maps and optic fiber tracts from datasets used in Figure 1A. We found that virus expression and optic fiber tracts were located in the ventrolateral POA, lateral POA, and the lateral part of medial POA, and included this description in the text.

**Author response image 1. sa3fig1:** Location of virus expression (A) and optic fiber placement (B) within subregions of POA. Mouse brain figure adapted from the Allen Reference Atlas - Mouse Brain.

(5) It would be ideal to further characterize these particular GAD2 cells by RT-PCR or RNA seq. Which other markers do they express?

Single-cell RNA-sequencing of POA neurons has revealed an enormous level of molecular diversity, consisting of nearly 70 subpopulations based on gene expression of which 43 can be clustered into inhibitory neurons (Moffitt et al., 2018) . One of the most studied subpopulation of POA sleep-active neurons contains the inhibitory neuropeptide galanin (Sherin et al., 1998; Gaus et al., 2002; Chung et al., 2017; Kroeger et al., 2018; Ma et al., 2019; Miracca et al., 2022) . Galanin neurons have been demonstrated to innervate the TMN (Sherin et al., 1998) yet, within the galanin neurons 7 distinct clusters exist based on unique gene expression (Moffitt et al., 2018) . In addition to galanin, we have previously performed single-cell RNA-seq on POA GAD2 → TMN neurons and identified additional neuropeptides such as cholecystokinin (CCK), corticotropin-releasing hormone (CRH), prodynorphin (PDYN), and tachykinin 1 (TAC1) as subpopulations of GABAergic POA sleep-active neurons (Chung et al., 2017; Smith et al., 2023) . Like galanin, these neuropeptides can also be divided into multiple subtypes as well (Chen et al., 2017; Moffitt et al., 2018) . Thus while these molecular markers for POA neurons are immensely diverse, we agree that characterizing the molecular identity of POA GAD2 → TMN neurons and investigating the functional relevance of these neuropeptides in the context of REMs homeostasis would enrich our understanding of a neural circuit involved in REMs homeostasis and can stand as a separate extension of this manuscript.

**Reviewer #2 (Public Review):**
Maurer et al investigated the contribution of GAD2+ neurons in the preoptic area (POA), projecting to the tuberomammillary nucleus (TMN), to REM sleep regulation. They applied an elegant design to monitor and manipulate the activity of this specific group of neurons: a GAD2-Cre mouse, injected with retrograde AAV constructs in the TMN, thereby presumably only targeting GAD2+ cells projecting to the TMN. Using this set-up in combination with technically challenging techniques including EEG with photometry and REM sleep deprivation, the authors found that this cell-type studied becomes active shortly (≈40sec) prior to entering REM sleep and remains active during REM sleep. Moreover, optogenetic inhibition of GAD2+ cells inhibits REM sleep by a third and also impairs the rebound in REM sleep in the following hour. Despite a few reservations or details that would benefit from further clarification (outlined below), the data makes a convincing case for the role of GAD2+ neurons in the POA projecting to the TMN in REM sleep regulation.

We thank the reviewer for the thorough assessment of our study and supportive comments. We have addressed your concerns in the revised manuscript, and our point by point response is provided below.

The authors found that optogenetic inhibition of GAD2+ cells suppressed REM sleep in the hour following the inhibition (e.g. Fig2 and Fig4). If the authors have the data available, it would be important to include the subsequent hours in the rebound time (e.g. from ZT8.5 to ZT24) to test whether REM sleep rebound remains impaired, or recovers, albeit with a delay.

We thank the reviewer for this comment and agree that it would be interesting to know how REMs changes for a longer period of time throughout the rebound phase. For Fig. 2, we did not record the subsequent hours. For Fig 4, we recorded the subsequent rebound between ZT7.5 and 10.5. When we compare the REMs amount during this 4 hr interval, the SwiChR mice have less REMs compared with eYFP mice with marginal significance (unpaired t-test, p=0.0641). We also plotted the cumulative REMs amount during restriction and rebound phases, and found that the cumulative amount of REMs was still lower in SwiChR mice than eYFP mice at ZT 10.5 (Author response image 2). Therefore, it will be interesting to record for a longer period of time to test when the SwiChR mice compensate for all the REMs that was lost during the restriction period.

**Author response image 2. sa3fig2:** Cumulative amount of REMs during REMs deprivation and rebound combined with optogenetic stimulation in eYFP and SwiChR groups. This data is shown as bar graphs in Figure 4.

REM sleep is under tight circadian control (e.g. Wurts et al., 2000 in rats; Dijk, Czeisler 1995 in humans). To contextualize the results, it would be important to mention that it is not clear if the role of the manipulated neurons in REM sleep regulation hold at other circadian times of the day.

**Author response image 3. sa3fig3:** Inhibiting POA GAD2→ TMN neurons at ZT5-8 reduces REMs. (A) Schematic of optogenetic inhibition experiments. Mouse brain figure adapted from the Allen Reference Atlas - Mouse Brain. (B) Percentage of time spent in REMs, NREMs and wakefulness with laser in SwiChR++ and eYFP mice. Unpaired t-tests, p = 0.0013, 0.0469 for REMs and wakeamount. (C) Duration of REMs, NREMs, and wake episodes. Unpaired t-tests, p = 0.0113 for NREMs duration. (D) Frequency of REMs, NREMs, and wake episodes. Unpaired t-tests, p = 0.0063, 0.0382 for REMs and NREMs frequency.

REMs propensity is largest towards the end of the light phase (Czeisler et al., 1980; Dijk and Czeisler, 1995; Wurts and Edgar, 2000). As a control, we therefore performed the optogenetic inhibition experiments of POA GAD2→TMN neurons during ZT5-8 (Author response image 3). Similar to our results in Figure 2, we found that SwiChR-mediated inhibition of POA GAD2 →TMN neurons attenuated REMs compared with eYFP laser sessions. These findings suggest our results are consistentat other circadian times of the day.

The effect size of the REM sleep deprivation using the vibrating motor method is unclear. In FigS4-D, the experimental mice reduce their REM sleep to 3% whereas the control mice spend 6% in REM sleep. In Fig4, mice are either subjected to REM sleep deprivation with the vibrating motor (controls), or REM sleep deprivations + optogenetics (experimental mice).The control mice (vibrating motor) in Fig4 spend 6% of their time in REM sleep, which is double the amount of REM sleep compared to the mice receiving the same treatment in FigS4-D. Can the authors clarify the origin of this difference in the text?

The effect size for REM sleep deprivation is now added in the text.

It is important to note that these figures are analyzing two different intervals of the REMs restriction. In Fig. S4D, we analyzed the total amount of REMs over the entire 6 hr restriction interval (ZT1.5-7.5). In Fig. 4, we analyzed the amount of REMs only during the last 3 hr of restriction (ZT4.5-7.5) as optogenetic inhibition was performed only during the last 3 hrs when the REMs pressure is high. In Fig. S4D, we looked at the amount of REMs during ZT1.5-4.5 and 4.5-7.5 and found that the amount of REMs during ZT4.5-7.5 (4.46 ± 0.25 %; mean ± s.e.m.) is indeed higher than ZT 1.5-4.5 (1.66 ± 0.62 %), and is comparable to the amount of REMs during ZT4.5-7.5 in eYFP mice (5.95 ± 0.52 %) in Fig. 4. We now clearly state in the manuscript at which time points we analyzed the amount, duration and frequency of REMs.

**Recommendations for the authors:**

**Reviewer #1 (Recommendations For The Authors):**
(1) A few further citations suggested: Discussion "The TMN contains histamine producing neurons and antagonizing histamine neurons causes sleepiness..." It would be appropriate to cite Uygun DS et al 2016 J Neurosci (PMID: 27807161) here. Using the same HDC-Cre mice as used by Maurer et al., Uygun et al found that selectively increasing GABAergic inhibition onto histamine neurons produced NREM sleep.

We apologize for omitting this important paper. In the revised manuscript, we added this citation.

(2) Materials and Methods.Although the JAX numbers are given for the mouse lines based on researchers generously donating to JAX for others to use, please cite the papers corresponding to the GAD2-ires-Cre and HDC-ires-Cre mouse lines deposited at JAX.GAD2-ires-Cre was described in Taniguchi H et al., 2011, Neuron (PMID: 21943598).The construction of the HDC-ires-CRE line is described in Zecharia AY et al J Neurosci et al 2012 (PMID: 22993424).

We have now added these important citations in the revised manuscript.

(3) Similarly, for the viruses, please provide the citations for the AAV constructs that were donated to Addgene.

We have now added these citations in the revised manuscript.

**Reviewer #2 (Recommendations For The Authors):**
The authors rely heavily on their conclusions by using an optogenetic tool that inhibits the activity of GAD2+ neurons, however, it is not shown that these neurons are indeed inhibited as expected. An alternative approach to tackle this could be the application of a different technique to achieve the same output (e.g. chemogenetics). However, both experiments (confirmation of inhibition, or using a different technique) would require a significant amount of work, and given the numerous studies out there showing that these optogenetic tools tend to work, may not be necessary. Hence the authors could also cite a similar study that used a likewise construct and where it was indeed shown that this technique works (i.e. similar retrograde optogenetic construct with Cre depedendent expression combined with electrophysiological recordings).

This laser stimulation protocol was designed based on previous reports of sustained inhibition using the same inhibitory opsin and our prior results that recapitulate similar findings as inhibitory chemogenetic techniques (Iyer et al., 2016; Kim et al., 2016; Wiegert et al., 2017; Stucynski et al., 2022). We have now added this description in the Result section.

Fig1A - Right: the virus expression graphs are great and give a helpful insight into the variability. The image on the left (GCAMP+ cells) is less clear, the GCAMP+ cells don't differentiate well from the background. Perhaps the whole brain image with inset in POA can show the GCAMP expression more convincingly.

We have added a histology picture showing the whole brain image with inset in the POA in the updated Fig. 1A .

Statistics: The table is very helpful. Based on the degrees of freedom, it seems that in some instances the stats are run on the recordings rather than on the individual mice (e.g. Fig1). It could be considered to use a mixed model where subjects as taken into account as a factor.

**Author response image 4. sa3fig4:** ΔF/Factivity of POA GAD2→TMN neurons during NREMs. The duration of NREMs episodes was normalized in time, ranging from 0 to 100%. Shading, ± s.e.m. Pairwise t-tests with Holm-Bonferroni correctionp = 5.34 e-4 between80 and100. Graybar, intervals where ΔF/F activity was significantly different from baseline (0 to 20%, the first time bin). n = 10 mice. In Fig. 1E , we ran stats based on the recordings. In this data set, we ran stats based on the individual mice, and found that the activity also gradually increased throughout NREMs episodes.

There is an effect of laser in Fig2 on REM sleep amount, as well as an interaction effect with virus injection (from the table). Therefore, it would be helpful for the reader to also show REM sleep data from the control group (laser stimulation but no active optogenetics construct) in Fig 2.

To properly control laser and virus effect, we performed the same laser stimulation experiments in eYFP control mice (expressing only eYFP without optogenetic construct, SwiChR++) and the data is provided in Fig2C .

Fig3B: At the start of the rebound of REM sleep, there is a massive amount of wakefulness, also reflected in the change of spectral composition. Could you comment on the text about what is happening here?

We quantified the amount of wakefulness during the first hour of REMs rebound and found that indeed there is no significant difference in wakefulness between REM restriction and baseline control conditions ( Fig. S4H ). Therefore, while the representative image in Fig 3B shows increased wakefulness at the beginning of REMs rebound, we do not think the overall amount of wakefulness is increased.

Fig 4, supplementary data: it would be helpful for the reader to have mentioned in the text the effect size of the REM sleep restriction protocol (e.g. mean and standard deviation).

Thank you for this suggestion. We have now added the effect size for the REM sleep restriction experiments in the main text.

REM sleep restriction and photometry experiment: could be improved by adding within the main body of text that, in order to conduct the photometry experiment in the last hours of REM sleep deprivation, the manual REM sleep deprivation had to be applied, because the vibrating motor technique disturbed the photometry recordings.

Thank you for this suggestion. We have added the description in the main text.

Suggestion to build further on the already existing data (not for this paper): you have a powerful dataset to test whether REM sleep pressure builds up during wakefulness or NREM sleep, by correlating when your optogenetic treatment occurs (NREM or wakefulness), with the subsequent rebound in REM sleep (see also Endo et al., 1998; Benington and Heller, 1994; Franken 2001).

We thank the reviewer for this excellent suggestion. We plan to carry out this experiment in the future.

References

Antila, H., Kwak, I., Choi, A., Pisciotti, A., Covarrubias, I., Baik, J., et al. (2022). A noradrenergic-hypothalamic neural substrate for stress-induced sleep disturbances. Proc. Natl. Acad. Sci. 119, e2123528119. doi: 10.1073/pnas.2123528119.

Blake, H., and Gerard, R. W. (1937). Brain potentials during sleep. Am. J. Physiol.-Leg. Content 119, 692–703. doi: 10.1152/ajplegacy.1937.119.4.692.

Chen, R., Wu, X., Jiang, L., and Zhang, Y. (2017). Single-Cell RNA-Seq Reveals Hypothalamic Cell Diversity. Cell Rep. 18, 3227–3241. doi: 10.1016/j.celrep.2017.03.004.

Chung, S., Weber, F., Zhong, P., Tan, C. L., Nguyen, T., Beier, K. T., et al. (2017). Identification of Preoptic Sleep Neurons Using Retrograde Labeling and Gene Profiling. Nature 545, 477–481. doi: 10.1038/nature22350.

Czeisler, C. A., Zimmerman, J. C., Ronda, J. M., Moore-Ede, M. C., and Weitzman, E. D. (1980). Timing of REM sleep is coupled to the circadian rhythm of body temperature in man. Sleep 2, 329–346.

Dijk, D. J., Brunner, D. P., Beersma, D. G., and Borbély, A. A. (1990). Electroencephalogram power density and slow wave sleep as a function of prior waking and circadian phase. Sleep 13, 430–440. doi: 10.1093/sleep/13.5.430.

Dijk, D. J., and Czeisler, C. A. (1995). Contribution of the circadian pacemaker and the sleep homeostat to sleep propensity, sleep structure, electroencephalographic slow waves, and sleep spindle activity in humans. J. Neurosci. Off. J. Soc. Neurosci. 15, 3526–3538. doi: 10.1523/JNEUROSCI.15-05-03526.1995.

Donlea, J. M., Pimentel, D., and Miesenböck, G. (2014). Neuronal machinery of sleep homeostasis in *Drosophila*. Neuron 81, 860–872. doi: 10.1016/j.neuron.2013.12.013.

Ferrara, M., De Gennaro, L., Casagrande, M., and Bertini, M. (1999). Auditory arousal thresholds after selective slow-wave sleep deprivation. Clin. Neurophysiol. Off. J. Int. Fed. Clin. Neurophysiol. 110, 2148–2152. doi: 10.1016/s1388-2457(99)00171-6.

Gaus, S. E., Strecker, R. E., Tate, B. A., Parker, R. A., and Saper, C. B. (2002). Ventrolateral preoptic nucleus contains sleep-active, galaninergic neurons in multiple mammalian species. Neuroscience 115, 285–294. doi: 10.1016/S0306-4522(02)00308-1.

Gong, H., McGinty, D., Guzman-Marin, R., Chew, K.-T., Stewart, D., and Szymusiak, R. (2004). Activation of c-fos in GABAergic neurones in the preoptic area during sleep and in response to sleep deprivation. J. Physiol. 556, 935–946. doi: 10.1113/jphysiol.2003.056622.

Iyer, S. M., Vesuna, S., Ramakrishnan, C., Huynh, K., Young, S., Berndt, A., et al. (2016). Optogenetic and chemogenetic strategies for sustained inhibition of pain. Sci. Rep. 6, 30570. doi: 10.1038/srep30570.

Kim, H., Ährlund-Richter, S., Wang, X., Deisseroth, K., and Carlén, M. (2016). Prefrontal Parvalbumin Neurons in Control of Attention. Cell 164, 208–218. doi: 10.1016/j.cell.2015.11.038.

Kroeger, D., Absi, G., Gagliardi, C., Bandaru, S. S., Madara, J. C., Ferrari, L. L., et al. (2018). Galanin neurons in the ventrolateral preoptic area promote sleep and heat loss in mice. Nat. Commun. 9, 4129. doi: 10.1038/s41467-018-06590-7.

Ma, Y., Miracca, G., Yu, X., Harding, E. C., Miao, A., Yustos, R., et al. (2019). Galanin Neurons Unite Sleep Homeostasis and α2-Adrenergic Sedation. Curr. Biol. CB 29, 3315-3322.e3. doi: 10.1016/j.cub.2019.07.087.

Mallick, B. N., and Singh, A. (2011). REM sleep loss increases brain excitability: role of noradrenaline and its mechanism of action. Sleep Med. Rev. 15, 165–178. doi: 10.1016/j.smrv.2010.11.001.

McDermott, C. M., LaHoste, G. J., Chen, C., Musto, A., Bazan, N. G., and Magee, J. C. (2003). Sleep deprivation causes behavioral, synaptic, and membrane excitability alterations in hippocampal neurons. J. Neurosci. Off. J. Soc. Neurosci. 23, 9687–9695. doi: 10.1523/JNEUROSCI.23-29-09687.2003.

Miracca, G., Anuncibay-Soto, B., Tossell, K., Yustos, R., Vyssotski, A. L., Franks, N. P., et al. (2022). NMDA Receptors in the Lateral Preoptic Hypothalamus Are Essential for Sustaining NREM and REM Sleep. J. Neurosci. 42, 5389–5409. doi: 10.1523/JNEUROSCI.0350-21.2022.

Moffitt, J. R., Bambah-Mukku, D., Eichhorn, S. W., Vaughn, E., Shekhar, K., Perez, J. D., et al. (2018). Molecular, spatial, and functional single-cell profiling of the hypothalamic preoptic region. Science 362. doi: 10.1126/science.aau5324.

Neckelmann, D., and Ursin, R. (1993). Sleep stages and EEG power spectrum in relation to acoustical stimulus arousal threshold in the rat. Sleep 16, 467–477.

Park, S.-H., Baik, J., Hong, J., Antila, H., Kurland, B., Chung, S., et al. (2021). A probabilistic model for the ultradian timing of REM sleep in mice. PLOS Comput. Biol. 17, e1009316. doi: 10.1371/journal.pcbi.1009316.

Rosa, R. R., and Bonnet, M. H. (1985). Sleep stages, auditory arousal threshold, and body temperature as predictors of behavior upon awakening. Int. J. Neurosci. 27, 73–83. doi: 10.3109/00207458509149136.

Sherin, J. E., Elmquist, J. K., Torrealba, F., and Saper, C. B. (1998). Innervation of histaminergic tuberomammillary neurons by GABAergic and galaninergic neurons in the ventrolateral preoptic nucleus of the rat. J. Neurosci. Off. J. Soc. Neurosci. 18, 4705–4721.

Smith, J., Honig-Frand, A., Antila, H., Choi, A., Kim, H., Beier, K. T., et al. (2023). Regulation of stress-induced sleep fragmentation by preoptic glutamatergic neurons. Curr. Biol. CB , S0960-9822(23)01585–3. doi: 10.1016/j.cub.2023.11.035.

Stucynski, J. A., Schott, A. L., Baik, J., Chung, S., and Weber, F. (2022). Regulation of REM sleep by inhibitory neurons in the dorsomedial medulla. Curr. Biol. CB 32, 37-50.e6. doi: 10.1016/j.cub.2021.10.030.

Takahashi, K., Lin, J.-S., and Sakai, K. (2009). Characterization and mapping of sleep-waking specific neurons in the basal forebrain and preoptic hypothalamus in mice. Neuroscience 161, 269–292. doi: 10.1016/j.neuroscience.2009.02.075.

Weber, F., Hoang Do, J. P., Chung, S., Beier, K. T., Bikov, M., Saffari Doost, M., et al. (2018). Regulation of REM and Non-REM sleep by periaqueductal GABAergic neurons. Nat. Commun. 9, 1–13. doi: 10.1038/s41467-017-02765-w.

Wiegert, J. S., Mahn, M., Prigge, M., Printz, Y., and Yizhar, O. (2017). Silencing Neurons: Tools, Applications, and Experimental Constraints. Neuron 95, 504–529. doi: 10.1016/j.neuron.2017.06.050.

Williams, H. L., Hammack, J. T., Daly, R. L., Dement, W. C., and Lubin, A. (1964). RESPONSES TO AUDITORY STIMULATION, SLEEP LOSS AND THE EEG STAGES OF SLEEP. Electroencephalogr. Clin. Neurophysiol. 16, 269–279. doi: 10.1016/0013-4694(64)90109-9.

Wurts, S. W., and Edgar, D. M. (2000). Circadian and homeostatic control of rapid eye movement (REM) sleep: promotion of REM tendency by the suprachiasmatic nucleus. J. Neurosci. Off. J. Soc. Neurosci. 20, 4300–4310. doi: 10.1523/JNEUROSCI.20-11-04300.2000.

Zhou, Y., Lai, C. S. W., Bai, Y., Li, W., Zhao, R., Yang, G., et al. (2020). REM sleep promotes experience-dependent dendritic spine elimination in the mouse cortex. Nat. Commun. 11, 4819. doi: 10.1038/s41467-020-18592-5.